# Improved Guarantees for Fully Dynamic $k$-Center Clustering with Outliers in General Metric Spaces

**Leyla Biabani**
Eindhoven University of Technology
Eindhoven, The Netherlands
l.biabani@tue.nl

**Annika Hennes**
Heinrich Heine University Düsseldorf
Düsseldorf, Germany
annika.hennes@hhu.de

**Denise La Gordt Dillie**
Eindhoven University of Technology
Eindhoven, The Netherlands
lagordtdilliedenise@gmail.com

**Morteza Monemizadeh**
Eindhoven University of Technology
Eindhoven, The Netherlands
m.monemizadeh@tue.nl

**Melanie Schmidt**
Heinrich Heine University Düsseldorf
Düsseldorf, Germany
mschmidt@hhu.de

## Abstract

The metric $k$-center clustering problem with $z$ outliers, also known as $(k, z)$-center clustering, involves clustering a given point set $P$ in a metric space $(M, d)$ using at most $k$ balls, minimizing the maximum ball radius while excluding up to $z$ points from the clustering. This problem holds fundamental significance in various domains, such as machine learning, data mining, and database systems.

This paper addresses the fully dynamic version of the problem, where the point set undergoes continuous updates (insertions and deletions) over time. The objective is to maintain an approximate $(k, z)$-center clustering with efficient update times. We propose a novel fully dynamic algorithm that maintains a $(4 + \epsilon)$-approximate solution to the $(k, z)$-center clustering problem that covers all but at most $(1 + \epsilon)z$ points at any time in the sequence with probability $1 - k/e^{\Omega(\log k)}$. The algorithm achieves an expected amortized update time of $\mathcal{O}(\epsilon^{-3}k^6 \log(k) \log(\Delta))$, and is applicable to general metric spaces. Our dynamic algorithm presents a significant improvement over the recent dynamic $(14 + \epsilon)$-approximation algorithm by Chan, Lattanzi, Sozio, and Wang [5] for this problem.

## 1 Introduction

Clustering problems and algorithms play an important role across a multitude of fields, helping researchers and practitioners in the analysis of data and identification of patterns. These techniques find extensive application in diverse domains, including machine learning, where they help in categorizing and understanding complex datasets. In data mining, clustering methods are utilized to uncover hidden structures and relationships within large datasets, facilitating better decision-making and insight generation. Moreover, in image and signal processing, clustering algorithms assist in segmenting and classifying data, enabling tasks such as image recognition and signal denoising.

In bioinformatics, clustering techniques are essential for organizing biological data and identifying patterns in genetic sequences, protein structures, and gene expression profiles. Similarly, in anomaly

38th Conference on Neural Information Processing Systems (NeurIPS 2024).

detection, clustering methods are employed to identify unusual or unexpected patterns in data, which may indicate potential anomalies or security breaches. Furthermore, in social network analysis, clustering algorithms help in understanding the structure and dynamics of social networks by identifying communities and influential nodes.

The $k$-center problem is known as one of the fundamental clustering problems. Given a set of points $P$ in a metric space and a number $k$, the aim of the $k$-center problem is to find $k$ centers such that the maximum distance between any point and its closest center is minimized. This can also be equivalently formulated as finding a minimum radius $r$ and centers $c_1, \ldots, c_k$ such that the balls $\cup_{i=1}^{k} \mathcal{B}_P(c_i, r)$ cover the point set. The $k$-center problem can be 2-approximated, and this is the best possible approximation guarantee [13]. In the last decade, the focus has shifted to analyzing the problem under various complications that arise in applications.

One line of research is to study the $k$-center problem (and other clustering problems) in different computational models like *streaming* or for *dynamic* point sets. In the first case, points arrive sequentially, and only a summary can be stored in memory. In the second case, the point set is maintained by insertion queries and deletion queries for single points, and algorithms have to update their solution after any such query.

Another line of research is to study clustering under constraints. For example, *capacitated* clustering is very popular, i.e., limiting the number of points per cluster. However, lower bounds on cluster sizes have also been studied in the context of anonymity, and newer works have also considered constraints that model societal concerns like fair or diverse composition of clusters. In this paper, we study clustering *with outliers*. Formulated as a constraint, the $k$-center problem with outliers allows $k + z$ centers, but $z$ of these have to be singletons, meaning no point may be assigned to them. We can intuitively formulate it with balls as follows: The $k$-center problem with outliers asks to find a minimum $r$ and $k$ centers $c_1, \ldots, c_k$ such that the balls $\cup_{i=1}^{k} \mathcal{B}_P(c_i, r)$ cover all but $z$ points.

Solving clustering problems in the presence of outliers is a crucial task due to the common occurrence of measurement errors or other sources of significant deviation from the rest of the data in real-world datasets. Ignoring outliers can severely distort the results of clustering algorithms, leading to inaccurate groupings. To address this challenge, a common approach is to solve a clustering problem while excluding up to $z$ data points considered as outliers.

The first approximation algorithm for the $k$-center problem with outliers is due to Charikar et al. [8]. The challenge when designing approximation algorithms in the presence of outliers is that one needs to show that *enough* points are covered by balls of bounded sizes around the approximate centers. It is not necessary to identify the outliers of an optimal solution *exactly* as long as the number of uncovered points remains small enough. Due to this, [8] and follow-up papers use *charging* arguments. Points covered by the solution of the algorithm are mapped to points in optimum clusters, and it is then shown that the number of uncharged points is small enough.

We explore the $k$-center clustering problem with $z$ outliers within the fully dynamic model, where the point set experiences continuous updates through insertions and deletions over time. Quite some work on this problem has been done for inputs from metric spaces with *bounded doubling dimension* [2, 3, 19]. This setting allows for geometric data structures that allow easier navigation through the data set and also bounding of the number of changes that can occur due to a query by dimension-dependent volume arguments.

We study the general metric setting. General metric spaces represent a key objective for clustering algorithms due to their broad range of distance functions, ensuring applicability to any data type. Chan et al. recently showed that in general metric spaces, any dynamic $\mathcal{O}(1)$-approximation algorithm for $k$-center clustering excluding at most $z$ outliers has an amortized update time of $\Omega(z)$ [5]. For real-world applications, the fraction of outliers of the data could be arbitrarily large. Therefore, we allow for $(1 + \epsilon)z$ outliers to be excluded to avoid the dependence on $z$ of the update time.

The only previous work in this setting is the algorithm by Chan et al. [5], which returns a solution with at most $(1 + \epsilon)z$ outliers with probability at least $1 - \delta$, for $0 < \delta \leq \frac{1}{k}$ and $\epsilon > 0$. It works by maintaining a clustering with $t := k \cdot \lceil \log_{1+\epsilon} \frac{k}{\delta} \rceil$ clusters of radius $2r$ and using this to derive a final clustering with $k$ clusters and radius $14r$. They maintain such a clustering for all $r \in \Gamma := \{(1 + \tau)^i : d_{\min} \leq (1 + \tau)^i \leq (1 + \tau) \cdot d_{\max}, i \in \mathbb{N}\}$, with $d_{\min}$ and $d_{\max}$ the minimum and maximum distances, respectively, between any two points ever inserted. It is then shown that there exists an instance $r \in \Gamma$ that will approximate the optimal radius to within a factor $(14 + \tau)$,

while allowing for $(1 + \epsilon)z$ outliers, with probability at least $1 - \delta$. The amortized time per update is $\mathcal{O}(|\Gamma| \cdot \frac{k^2}{\epsilon^2} \log^2 \frac{1}{\delta})$, with $|\Gamma| = (\log \frac{d_{\max}}{d_{\min}})/\tau$. The total memory requirement is $\mathcal{O}(|\Gamma| \cdot |n|)$ where $n$ is the number of points in the current set.

## 1.1 Our contribution

We introduce a novel fully dynamic $(4 + \epsilon)$-approximation algorithm designed to maintain a $k$-center clustering while allowing for at most $(1 + \epsilon)z$ outliers at any time in the sequence.

The expected amortized update time is $\mathcal{O}(\epsilon^{-3} k^6 \log(k) \log(\Delta))$ per operation (insertion or deletion). This is independent of $z$ and of the number of points currently present in the sequence. This characteristic makes the algorithm applicable to real-world scenarios.

Notice that our data structure is continually storing an actual solution, so we can produce this solution at any time and do not need to specify additional query times (like, for example [9, 2, 18]).

Our main technical contribution is a novel combination of the sampling-based level data structure by Chan et al. [5] and the original greedy strategy by Charikar et al. [8] that enables us to achieve a much improved approximation guarantee compared to [5].

**Theorem 1.1.** *Let $(M, d)$ be a metric space and $\epsilon > 0$ be an accuracy parameter. The spread ratio $\Delta = \frac{d_{\max}}{d_{\min}}$ of all points ever inserted is assumed to be bounded. There exists a randomized fully dynamic algorithm that maintains a $k$-center solution that allows up to $(1 + \epsilon)z$ many outliers on the current set of points. At every point in time, the current clustering is a $(4 + \epsilon)$-approximation to an optimal solution for the $(k, z)$-center problem with high probability. Upon insertion or deletion of a point, the data structure is updated in amortized update time $\mathcal{O}(\epsilon^{-3} k^6 \log(k) \log(\Delta))$.*

To achieve this, we make use of a data structure that is described in Section 2.1 and maintained by the respective algorithms handling updates to the point set. The algorithm handling insertions is specified in Section 2.2, and the deletion algorithm is described in Section 2.3. Similar to the approach by Chan et al. [5], we maintain a hierarchical structure consisting of levels, each containing one cluster. Unlike their work, however, we only maintain $\leq k$ levels. For the correct radius guess, the union of these levels directly gives the desired $(4 + \epsilon)$-approximation. If a level violates certain properties, this and all higher levels are reclustered. As opposed to [5], we do not need to recluster every time a center gets deleted but only when a cluster does not contain enough points anymore. The algorithm handling the reclustering is described in Section 2.4.

Our analysis of the approximation ratio follows a similar argument as the proof of the 3-approximation for static $k$-center with $z$ outliers as given by Charikar et al. [8] and works by charging points from chosen clusters to points in optimal clusters. We follow the same iterative charging method but extend their approach by maintaining a set of artificial outliers in order to adjust the argument to our algorithm specifically.

Note that we assume that a center can be any point from the underlying metric space, if the point was present in the data set once. Requiring that centers can only be placed at currently active points, we can achieve a 6-approximation. We formalize this in Lemma G.1.

## 1.2 Further Related work

The $k$-center problem is very well understood; two 2-approximation algorithms for it are known [13, 15] and a matching lower bound is also known [16]. The $k$-center problem with outliers was first studied and 3-approximated in [8]. In 2016, [4] gave a 2-approximation for this problem, but the algorithm is rather complex and less amenable to practical implementation. Charikar et al. [7] developed an elegant algorithm to maintain an 8-approximation for the vanilla $k$-center problem in the streaming model. Streaming can be seen as a dynamic insertion-only model. McCutchen and Khuller [17] improved this algorithm to a $(2 + \epsilon)$-approximation and also extended it to a $(4 + \epsilon)$-approximation for $k$-center with outliers. The $k$-center problem with outliers has also been studied in the sliding window model [10, 18] that also bears some similarity with our setting. For bounded doubling dimension, [18] gave a $(3 + \epsilon)$-approximation, but the algorithm for the general metric case is only stated as a $\mathcal{O}(1)$-approximation.

The fully dynamic $k$-center problem without outliers in general metrics was studied by [1, 6, 11], and the best-known result is a $(2+\epsilon)$-approximation with an amortized update time of $\mathcal{O}(k \operatorname{polylog}(n, \Delta))$ by Bateni et al. [1]. The same paper also shows that any algorithm that provably satisfies a non-trivial approximation guarantee needs $\Omega(nk)$ queries to the distance function, i.e., the amortized update time is in $\Omega(k)$, making their algorithm close to tight. The fully dynamic $k$-center problem without outliers was also studied for metrics with bounded doubling dimension [14, 19, 12] and in Euclidean space [20].

## 1.3 Preliminaries

In the preliminaries section, we provide an introduction to the $(k, z)$-center problem and discuss the dynamic model used in our paper.

We study the $k$-center clustering problem with $z$ outliers, formally defined as follows.

**Definition 1.2** ($(k, z)$-center clustering). Let $P$ be a point set in a metric space $(M, d)$ and let $k, z \in \mathbb{N}$ be two parameters. The goal is to compute a set $C \subseteq M$ of size at most $k$, such that the maximum distance of all but at most $z$ points to their nearest $c \in C$ is minimized. That is, find $C$ such that $\min_{Z \subseteq P, |Z| \leq z} \max_{x \in P \setminus Z} \min_{c \in C} d(x, c)$, with $|C| \leq k$.

We define $d_{\min}$ and $d_{\max}$ as the minimum and maximum distances, respectively, between any two points ever inserted. The ratio $\Delta = \frac{d_{\max}}{d_{\min}}$ is referred to as the spread ratio, and is assumed to be bounded. Let $r_{\text{OPT}}$ be the optimal radius for the $(k, z)$-center clustering problem of a point set $P$. We also define the ball $\mathcal{B}_P(c, r) = \{p \in P : d(c, p) \leq r\}$ to be the set of points in $P$ that are within distance $r$ from $c$. If $P$ is clear from the context, we may drop $P$ from the definition $\mathcal{B}_P(c, r)$. For a non-negative integer $m$, we denote $\{1, \cdots, m\}$ by $[m]$.

In the version of the $(k, z)$-center clustering problem that we consider, we allow $(1 + \epsilon)z$ outliers instead of strictly $z$. Furthermore, we require centers of clusters to be in the metric space $(M, d)$, but they need not be currently present in $P$. This is referred to as the non-discrete version of the problem. Lemma G.1 in the Appendix shows how our data structure can also support the discrete version of the problem and provides a 6-approximation solution for it.

**Fully dynamic model.** We consider the $(k, z)$-center clustering problem in the *fully dynamic model* against an *adaptive adversary*. In this model, we start with an empty point set, $P = \emptyset$, and process a sequence of operations determined by the adversary. We assume the adversary does not know the random bits chosen by our algorithm; however, it can observe the algorithm's output and adapt its responses in real time (unlike an *oblivious adversary*, which fixes a sequence of operations in advance). Each operation can be either an insertion, where a point from the metric space $(M, d)$ is added to $P$, or a deletion, where a point currently in $P$ is removed. We assume only points currently in $P$ may be deleted. Let $P^t$ represent the point set $P$ after $t$ operations. In other words, $P^t$ consists of all points in $(M, d)$ that have been inserted but not deleted after $t$ operations.

## 2 Algorithm

To explain the main ideas of our algorithm, we start by describing an iterative offline algorithm, which gives a $(4 + \epsilon)$ approximation for the $(k, z)$-center clustering problem.

**Offline algorithm.** We start with a point set $P_1 = P$, $\epsilon > 0$ and parameters $k, z, r \in \mathbb{N}$. Here, $r$ is a fixed guess for the optimal radius. For each iteration $i$, we sample a set of $S_i \subset P_i$ of $|S_i| = \psi \epsilon^{-1} k^2 \log k$ points with replacement, uniformly at random. Here, $\psi \geq 6\beta$ is a constant, where $\beta > \alpha \geq 1$ are constants. We find a point $c_i \in S$, such that $\mathcal{B}(c_i, 2r)$ covers at least $\phi$ points from $P_i$, where $\phi$ is some threshold to be defined later. We then create a new cluster $C_i = \mathcal{B}_{P_i}(c_i, 4r)$, let $P_{i+1} = P_i \setminus C_i$, and continue to the next iteration. We stop once $k$ clusters have been created or $P_i = \emptyset$. Let $\lambda$ be a random variable representing the number of iterations we complete, where $\lambda$ can be at most $k$. After we have done $\lambda$ iterations we have computed clusters $C_1, C_2, ..., C_\lambda$ with corresponding centers $c_1, c_2, ..., c_\lambda$. The points that are not covered by the union of these clusters will be the set of outliers that our algorithm reports. If the set of outliers is at most $(1 + \epsilon)z$ points, we can report a solution for the $(k, z)$-center clustering problem. This offline algorithm will be used as a

sub-routine for the fully dynamic algorithm, which will be explained in Section 2.4. The definition of $\phi$ and the pseudocode of the algorithm will also be given in that section.

**Guesses for unknown $r_{\text{OPT}}$.** Since the optimal $r$ is usually not known, we can run the algorithm for all $r \in \mathcal{R} = \{(1+\epsilon)^i : d_{\min} \leq (1+\epsilon)^i \leq (1+\epsilon) \cdot d_{\max}, i \in \mathbb{N}\}$. We then choose the smallest $r \in \mathcal{R}$ such that all but at most $(1+\epsilon)z$ points are covered. We will show that this gives a $(4+\epsilon)$-approximation.

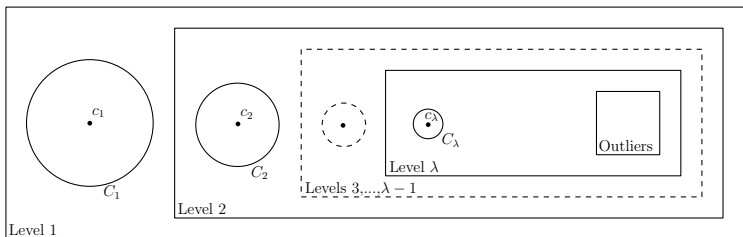

Figure 1: The $\lambda$ levels constructed by our offline algorithm.

**Leveling.** We can visualize the output of this algorithm as (at most) $\lambda + 1$ levels, where the first $\lambda \leq k$ levels each represent a cluster, and the last level represents the outliers (See Figure 1). More precisely, level $i$ represents cluster $C_i$ with center $c_i$, which was created in the $i^{th}$ iteration. Note that the construction of level $i$ only considers the points in $P_i$, which does not include points in the lower levels $\{1, 2, ..., i-1\}$.

The main idea for the fully dynamic algorithm is to maintain each level $i$ for $i \in [\lambda + 1]$ under an arbitrary number of insertions and deletions by updating the cluster $C_i$ when necessary. In the rest of the paper, we use subscript $i$ to refer to the level and superscript $t$ to refer to the time. For example, $P_i^t$ refers to the points present in level $i$ at time $t$, and $r_{\text{OPT}}^t$ refers to the optimal radius at time $t$. If the level or time is clear from the context, these may be omitted. For instance, if we define a situation at time $t'$, we will not repeat the superscript $t'$ on every term.

## 2.1 Data structure and invariants

We will construct a clustering $C_1, \ldots, C_\lambda$ of the current set of points $P^t$ that we store in the data structure $\mathcal{D}_r$ introduced by [6]. If $t$ is clear from the context, we will just write $P$ for $P^t$. As we do not know the current optimal radius, we will maintain one data structure $\mathcal{D}_r$ for every choice of $r \in \mathcal{R} = \{(1+\epsilon)^i : d_{\min} \leq (1+\epsilon)^i \leq d_{\max}, i \in \mathbb{N}\}$. Every data structure consists of up to $\lambda \leq k$ cluster-levels, with each level $i$ containing a cluster $C_i$. In level $i$, we keep track of the set of the points $P_i$ currently not covered, i.e., $P_i = P \setminus \bigcup_{j<i} C_j$. Level $\lambda + 1$ contains the outliers, i.e., $\mathcal{Z}_r = P \setminus \cup_{i \leq \lambda} C_i$. We denote $n_i = |P_i|$. By $\mathcal{B}(p, r)$ we denote the ball with radius $r$ centered at $p$, and $\mathcal{B}_A(p, r) = \mathcal{B}(p, r) \cap A$ for $A \subseteq P$.

Let $\alpha \geq 1$. The data structure $\mathcal{D}_r$ consists of the following components:

   ① A list $\mathcal{F}_r = \{c_1, c_2, ..., c_\lambda\}$ of $\lambda \leq k$ centers.

   ② A list $\mathcal{L}_r = \{C_1, C_2, ..., C_\lambda\}$ of clusters with $C_i = \mathcal{B}_{P_i}(c_i, 4r)$.

   ③ A set $\mathcal{Z}_r = P \setminus \left(\cup_{i \in [\lambda]} C_i\right)$ of outliers such that for all $i \in [\lambda]$ and $x \in \mathcal{Z}_r$, $d(x, c_i) > 4r$.

Let $\alpha \geq 1$ be a constant. We have the following invariants for $\mathcal{D}_r$.

---

   ① **Level invariant:** for all $i$, $P_{i+1} = P_i \setminus C_i$, with $C_i = \mathcal{B}_{P_i}(c_i, 4r)$.

   ② **Dense invariant:** for all $c_i \in \mathcal{F}_r$, $|\mathcal{B}_{P_i}(c_i, 2r)| \geq \min\left(z+1, \frac{n_i - z}{k-i+1} - \frac{\epsilon z}{\alpha k}\right)$.

---

## 2.2 Insertion

When a new point $p$ is inserted at time $t$, Procedure 3 in the appendix is executed. As input, we have $\mathcal{D}_r$ at time $t$, $p \in (M, d)$, $\epsilon > 0$ and $z, k \in \mathbb{N}$. First, we check whether $p$ is inside one of the existing clusters, in which case we can add $p$ to such a cluster. Otherwise, we add $p$ to $\mathcal{Z}_r$. Next, we check if the dense invariant is maintained after the insertion of the new point. Generally, the dense invariant can be broken in two ways:

❶ If the insertion of a new point results in more than $(1 + \epsilon)z$ outliers, then Lemma 3.7 shows that the dense invariant is no longer valid at some level $i$.

❷ If the addition of a point $p$ to a cluster $C_i$ causes the dense invariant to be violated at some lower level $j < i$, that occurs because $n_j^{t+1} = n_j^t + 1$.

If the dense invariant is broken for some level $i$, we recluster levels $i, ..., \lambda$ with $\lambda \leq k$ by invoking Procedure 5 as a sub-routine. Observe that if there are multiple levels $i$ where the dense invariant is broken, we choose the lowest one.

## 2.3 Deletion

For the deletion of a point $p$ at time $t$, Procedure 4 in the appendix is executed. The input is $\mathcal{D}_r$ at time $t$, $p \in (M, d)$, $\epsilon > 0$ and $z, k \in \mathbb{N}$. First we check if $p$ is an outlier, in which case we remove $p$ from $\mathcal{Z}_r$. If $p$ is either a center or a point in a cluster, we find cluster $C_i$ which contains $p$. If cluster $C_i = \mathcal{B}(c_i, 4r)$ contains at least $\min\left(z + 1, \frac{|P_i| - z}{k - i + 1} - \frac{\epsilon z}{\alpha k}\right)$ points, we do not re-cluster and simply remove $p$ from $C_i$. Note that the underlying point set exists in the metric space $(M, d)$. If the center $c_i$ of a cluster $C_i$ is deleted, provided that the dense invariant remains valid, we can continue to utilize $c_i$ as the center of cluster $C_i$, given our knowledge that the point $c_i$ is located within the metric space $(M, d)$. In Lemma G.1, we explain how we can still obtain a 6-approximation if centers have to come from the current point set. If $|C_i| < \min\left(z + 1, \frac{|P_i| - z}{k - i + 1} - \frac{\epsilon z}{\alpha k}\right)$ after the deletion of $p$, then it also follows that $|\mathcal{B}(c_i, 2r) \cap P_i| < \min\left(z + 1, \frac{|P_i| - z}{k - i + 1} - \frac{\epsilon z}{\alpha k}\right)$, i.e., the dense invariant is violated on this level. In this case, we want to redistribute the points in $P_i$ such that the levels $i, \dots, \lambda$ fulfill the dense invariant. Deletion of a point in level $i$ does not violate the invariants at levels $1, \dots, i - 1$. We re-cluster the points in $(\cup_{i \leq j \leq k} C_j) \cup \mathcal{Z}_r$ using Procedure 5. We finish by updating $\mathcal{D}_r$.

## 2.4 Clustering sub-routine

The clustering sub-routine is the offline algorithm that was described at the beginning of Section 2. The pseudocode of this sub-routine is shown in Procedure 5. We use it to iteratively build the levels of data structure $\mathcal{D}_r$. Two cases are distinguished based on whether $z$ is small or large compared to the number of points in level $i$. In line 4, $\psi$ is a constant with $\psi \geq 6\beta$, where $\beta > \alpha$, and $\alpha$ is the constant used in the dense invariant. The threshold $\phi$ that was introduced in Section 2 is different depending on the size of $z$ compared to $n_i$. If $z + 1 \leq \frac{n_i - z}{4(k - i + 1)}$, then $\phi = \frac{n_i - z}{2(k - i + 1)}$ and if $z + 1 > \frac{n_i - z}{4(k - i + 1)}$, then $\phi = \frac{n_i - z}{k - i + 1} - \frac{\epsilon z}{\beta k}$. If we find multiple points $p^*$ in line 6 or 12 we choose one arbitrarily.

## 2.5 All aspects combined

The first step will be to initialize all $\mathcal{D}_r$ such that $\mathcal{F}_r, \mathcal{L}_r, \mathcal{Z}_r = \emptyset$. Then, the algorithm waits for an insertion or deletion operation. If a point is inserted, Procedure 3 is executed, and if a point is deleted, Procedure 4 is executed. The algorithm is ran simultaneously for all $r \in \mathcal{R} = \{(1 + \epsilon)^i : d_{\min} \leq (1 + \epsilon)^i \leq (1 + \epsilon) \cdot d_{\max}, i \in \mathbb{N}\}$.

In the next sections, it will be shown that a $(4 + \epsilon)$-approximation is given by the clustering $\mathcal{L}_r$ with $r \in \mathcal{R}$ the smallest $r$ for which $|\mathcal{Z}_r| \leq (1 + \epsilon)z$.

# 3 Analysis

We begin our analysis of the dynamic algorithm by introducing the concept of a dense cluster and specifying the criteria for a valid solution.

**Definition 3.1** (Dense cluster). A cluster $\mathcal{B}_{P_i}(c_i, 4r)$ is dense with respect to point set $P_i$ if it satisfies the dense invariant. That is, $|\mathcal{B}_{P_i}(c_i, 2r)| \geq \min\left(z + 1, \frac{n_i - z}{k - i + 1} - \frac{\epsilon z}{\alpha k}\right)$.

Observe that for the dense invariant, we consider the ball $\mathcal{B}_{P_i}(c_i, 2r)$ with a radius of $2r$. However, a cluster (as seen in line 18 of Procedure5) corresponds to the points within the ball $\mathcal{B}_{P_i}(c_i, 4r)$, which has a radius of $4r$.

**Definition 3.2** (Valid solution). A solution $\mathcal{B}_{P_1}(c_i, 4r) \cup \cdots \cup \mathcal{B}_{P_\lambda}(c_\lambda, 4r)$ with $\lambda \leq k$ for point set $P_1$ is valid if it covers all but at most $(1 + \epsilon)z$ points from $P_1$. We refer to each cluster $\mathcal{B}_{P_j}(c_j, 4r)$ for $j \in \{i, \ldots, \lambda\}$ of a valid solution as a valid cluster.

**Lemma 3.3** (Running time offline algorithm). *The running time of the offline algorithm, shown in Procedure 5, is $\mathcal{O}(n\epsilon^{-1}k^3 \log(k))$, where $n = |P|$.*

*Proof.* In each iteration of the while-loop, we need to compute $|\mathcal{B}(p, 2r)|$ for all $p \in S_i$. Since $|S_i| = \mathcal{O}(\epsilon^{-1}k^2 \log k)$, this takes $\mathcal{O}(n\epsilon^{-1}k^2 \log k)$ time. There can be at most $k$ iterations of the while-loop and hence the total running time is $\mathcal{O}(n\epsilon^{-1}k^3 \log k)$. $\square$

## 3.1 Maintaining invariants

In Supplementary Section B, we prove the following three lemmas. Lemmas 3.4 and 3.5 show that the dense and level invariants are maintained during the insertion or deletion of a point, respectively. Additionally, Lemma 3.6 shows that both invariants are maintained when invoking Procedure 5 as a subroutine upon the insertion or deletion of an arbitrary point with high probability if $r \geq r_{\text{OPT}}$. The case where $r < r_{\text{OPT}}$ is considered in Lemmas E.1 and E.2.

**Lemma 3.4** (Procedure 3 maintains invariants). *Assume that at time $t$, we have point set $P^t$, data structure $\mathcal{D}_r = (\mathcal{F}_r, \mathcal{L}_r, \mathcal{Z}_r)$. We assume that the level and dense invariants hold at time $t$ and $r \geq r_{\text{OPT}}^{t+1}$. At the start of time $t + 1$, we insert point $p$ using Procedure 3. After the insertion, the level and dense invariants still hold with probability 1 if Procedure 5 was not called and with the probability of at least $1 - \frac{2(k-i+1)}{e^{\Psi \log k}}$, where $\Psi \geq 1$ if Procedure 5 was not called.*

**Lemma 3.5** (Procedure 4 maintains invariants). *Assume that at time $t$, we have point set $P^t$, instance $\mathcal{D}_r = (\mathcal{F}_r, \mathcal{L}_r, \mathcal{Z}_r)$, parameters $k, z \in \mathbb{N}$ and $\epsilon > 0$. We assume that the level and dense invariants hold at time $t$ and $r \geq r_{\text{OPT}}^{t+1}$. At the start of time $t + 1$, we delete an arbitrary point $p$ using Procedure 4. After the deletion, the level and dense invariants hold with probability 1 if Procedure 5 was not called, and with probability $1 - \frac{2(k-i+1)}{e^{\Psi \log k}}$ with $\Psi \geq 1$ if Procedure 5 was called.*

**Lemma 3.6** (Procedure 5 maintains invariants with high probability). *Suppose the level and dense invariants hold for all levels $j < i$ and we call Procedure 5 on $P_i$ as the result of an insertion or deletion. Let $\lambda \leq k$ be a random variable representing the number of levels we have after completing Procedure 5. If $r \geq r_{\text{OPT}}$, Procedure 5 maintains the level and dense invariants for all levels $j$ with $i \leq j \leq \lambda$ with probability at least $1 - \frac{2(k-i+1)}{e^{\Psi \log k}}$, with $\Psi \geq 1$.*

## 3.2 Approximation guarantee

Next, we prove that if the invariants hold, our data structure $\mathcal{D}_r = (\mathcal{F}_r, \mathcal{L}_r, \mathcal{Z}_r)$ contains a valid solution to the $k$-center with $z$ outliers problem. Furthermore, if $r < (1 + \epsilon)r_{\text{OPT}}$, $\mathcal{D}_r$ provides a $(4 + \epsilon)$-approximation. Without loss of generality, for simplicity we assume that $\mathcal{D}_r$ contains $k$ levels. The proof of Lemma 3.7 follows the structure of the proof given in [8]. This proof uses the greediness of the algorithm to argue that the balls in the solution cover enough points to charge to. Here, we have to modify the proof to work with the dense invariant.

**Lemma 3.7** (Approximation guarantee). *Let $P$ be the current point set. Assume that the level invariant and the dense invariant hold for all levels $i \leq k$. Then we have the following guarantees:*

    ① **Valid solution:** *If $r_{OPT} \leq r$, then $\mathcal{B}(c_1, 4r) \cup \ldots \cup \mathcal{B}(c_k, 4r)$ covers all but at most $(1 + \frac{\epsilon}{\alpha})z$ outliers in $P$.*

    ② **Approximate solution:** *If $r_{OPT} \leq r < (1 + \epsilon)r_{OPT}$, then $\mathcal{B}(c_1, 4r) \cup \ldots \cup \mathcal{B}(c_k, 4r)$ gives a $(4 + \epsilon)$-approximation for the $k$-center with $z$ outliers problem on $P$.*

*Proof.* Recall that the level invariant states that for all $i$, we have $P_{i+1} = P_i \backslash \mathcal{B}(c_i, 4r)$ and the dense invariant states that for all $i$, we have that $|\mathcal{B}(c_i, 2r) \cap P_i| \geq \min(z + 1, \frac{n_i - z}{k - i + 1} - \frac{\epsilon z}{\alpha k})$.

Assume that in the optimal solution, we have balls $O_1, \ldots, O_k$ with radius $\leq r_{\text{OPT}}$. The union of these balls covers all but $z$ points in $P$. In order to prove Part ①, we aim to charge all but at most $\frac{\epsilon}{\alpha} z$ points of the optimal solution to points in our solution $\mathcal{B}(c_1, 4r) \cup \ldots \cup \mathcal{B}(c_k, 4r)$. We prove this by induction, and Part ② will then follow easily. In order to construct the charging argument, we need to argue that there are enough points in our solution to charge the points in the optimal balls. To this end, we construct modified optimal balls $O'_1, \ldots, O'_k$, where $O'_i \subseteq O_i$ for every $i \leq k$.

For the base case, we have $O'_1 = O_1, \cdots, O'_k = O_k$. We will show that we can order the modified balls in such a way, that at the end of time step $i$, all but at most $\frac{\epsilon z}{\alpha k} \cdot i$ points from the first $i$ modified balls are charged to distinct points in $\mathcal{B}(c_1, 4r) \cup \ldots \cup \mathcal{B}(c_i, 4r)$. This will allow us to prove that our solution covers at least as many points as the optimal solution.

Assume that all but at most $\frac{\epsilon z}{\alpha k} \cdot (i - 1)$ points in the first $i - 1$ modified balls $O'_1 \cup O'_2 \cup \ldots \cup O'_{i-1}$ have been charged to distinct points in $\mathcal{B}(c_1, 4r) \cup \ldots \cup \mathcal{B}(c_{i-1}, 4r)$ and consider iteration $i$. We distinguish two cases, namely if $\mathcal{B}(c_1, 2r) \cup \ldots \cup \mathcal{B}(c_i, 2r)$ intersects one of the remaining modified balls, or if it does not. The charging argument for each case proceeds as follows:

**Case 1.** Case 1 is when $\mathcal{B}(c_1, 2r) \cup \ldots \cup \mathcal{B}(c_i, 2r)$ intersects a remaining modified ball, call this ball $O'_i$. Note that $O'_i$ will be covered entirely by $\mathcal{B}(c_1, 4r) \cup \ldots \cup \mathcal{B}(c_i, 4r)$, since $r \geq r_{\text{OPT}}$. Hence, we charge the points of $O'_i$ to themselves and mark these points as covered. (See case 1 in Figure 2.) We call this charging *rule I*. Since the modified balls are disjoint [1], any point can be charged only once (to itself) by this rule. Next, we update $O'_1, O'_2, \ldots, O'_k$.

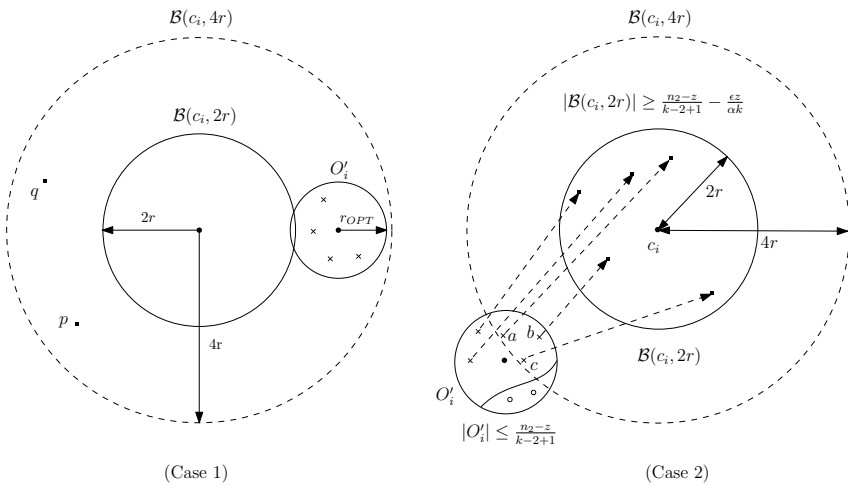

(Case 1)                    (Case 2)

Figure 2: Visualization of *charging rules* I and II. For case 1, we see that $\mathcal{B}(c_i, 2r)$ and $O_i$ intersect. As a result, $\mathcal{B}(c_i, 4r)$ covers all points in $O_i$. Points $p, q$ are in set $Z_i^c$. For case 2, the balls $\mathcal{B}(c_1, 2r)$ and $\mathcal{B}(c_2, 2r)$ do not intersect the optimal cluster $O'_i$. The crossed points in $O'_2$ are charged to black squared in $\mathcal{B}(c_2, 2r)$. The circle points in $O'_2$ are not charged to any point. Points $a, b, c$ are in $Z_i^d$.

We maintain two sets of credit points that we save and may use for future charging purposes. First, the set $Z_i^c = \mathcal{B}(c_i, 4r) \backslash (O'_1 \cup \ldots \cup O'_k)$ which is the set of points covered by $\mathcal{B}(c_i, 4r)$ that are not covered by $O'_1 \cup O'_2 \cup \ldots \cup O'_k$. We let $z_i^c = |Z_i^c|$ be the number of such points. In Figure 2, points $p, q$ are such points. Observe that no point is charged to points in $Z_i^c$, allowing us to use them later. We refer to these points as *credit points*.

There may also be previous modified balls $O'_j$, with $j < i$ that were considered in case 2 and are still present in $P_i$. More specifically, let $Z_i^d$ be the set of (there may exist) points in $O'_j \cap \mathcal{B}(c_j, 4r)$ that have been charged to distinct points in $\mathcal{B}(c_j, 4r)$ (See the discussion of case 2, below). For example, points $a, b, c$ in case 2 of Figure 2. Let $z_i^d = |Z_i^d|$ be the number of such points. Since no points

---

[1] In fact, original balls $O_1, \cdots, O_k$ can overlap, but we can make them disjoint by assigning any point that is inside multiple balls to just one of them arbitrarily.

are charged to points in $Z_i^d$, we save them as *credit points* for future charging purposes. We now update $O_1', O_2', \ldots, O_k'$ as follows: $O_1', \ldots, O_i'$ stay the same and we define $z_i^c + z_i^d$ artificial outliers in $(O_{i+1}' \cup \ldots \cup O_k') \cap P_{i+1}$.

**Case 2.** Case 2 occurs if $\mathcal{B}(c_1, 2r) \cup \ldots \cup \mathcal{B}(c_i, 2r)$ does not intersect any of the remaining modified balls. See Figure 2. Let $O_i'$ be one of the remaining modified balls covering $\leq \frac{n_i - z}{k - i + 1}$ points. Note that for finding $O_i'$ we do not count points that have been defined as artificial outliers, since these artificial outliers are already covered by balls of radius $4r$ in previous levels and we do not cover them by the remaining modified balls. We prove in Lemma 3.8 that such a ball $O_i'$ exists.

Using the assumption of this lemma that the dense invariant holds, we show in Lemma 3.9 that $|\mathcal{B}(c_i, 2r)| \geq \frac{n_i - z}{k - i + 1} - \frac{\epsilon z}{\alpha k}$. In this way, we find an upper bound for the number of points in $O_i'$ and a lower bound for the number of points in $\mathcal{B}(c_i, 2r)$. Now, we charge all but at most $\frac{\epsilon z}{\alpha k}$ points of $O_i'$ to the points of $\mathcal{B}(c_i, 2r)$ and mark these points as charged. We call this charging *rule II*. Note the difference between charging argument in Case 1 and 2. In Case 1, we charge the points in $O_i'$ to themselves, but in Case 2, we charge them to points covered by $\mathcal{B}(c_i, 2r)$. Recall that $\mathcal{B}(c_i, 2r)$ and $O_i'$ are disjoint.

Observe that points in balls $O_{i+1}', \cdots, O_k'$ will not be charged to points in $\mathcal{B}(c_i, 2r)$. Indeed, points in balls $O_{i+1}', \cdots, O_k'$ are charged to the balls that our algorithms find either using rule I or rule II. However, this will not happen based on rule I since $\mathcal{B}(c_i, 2r)$ is disjoint from all balls $O_{i+1}', \cdots, O_k'$. Moreover, rule II cannot also be applied since among balls $O_i', O_{i+1}', \cdots, O_k'$, we already charged the points in $O_i'$ to $\mathcal{B}(c_i, 2r)$ and points in $O_{i+1}', \cdots, O_k'$ will be charged to points in balls $\mathcal{B}(c_{i+1}, 2r), \cdots, \mathcal{B}(c_\lambda, 2r)$.

Next, we consider how we update $O_1', O_2', \ldots, O_k'$. Similar to case 1, we define the size of two sets of credits points. First, the variable $z_i^c = |\mathcal{B}(c_i, 4r) \backslash (O_1' \cup \ldots \cup O_k')|$ that corresponds to the number of points covered by $\mathcal{B}(c_i, 4r)$ that are not covered by $O_1' \cup O_2' \cup \ldots \cup O_k'$. There are $z_i^c - |O_i'|$ points from $\mathcal{B}(c_i, 4r)$ that are free (i.e., no point is charged to) and can still be charged. We consider these points as credits that we save and may use for future charging purposes.

There may also be previous modified balls $O_j'$, with $j < i$ that were considered in case 2 and are still present in $P_i$. More specifically, let $Z_i^d$ be the set of (there may exist) points in $O_j' \cap \mathcal{B}(c_j, 4r)$ that have been charged to distinct points in $\mathcal{B}(c_j, 4r)$. For example, points $a, b, c$ in case 2 of Figure 2. Let $z_i^d = |Z_i^d|$ be the number of such points. Since no points are charged to points in $Z_i^d$, we save them as *credit points* for future charging purposes. We now update $O_1', O_2', \ldots, O_k'$ as follows: $O_1', O_2', \ldots, O_i'$ stays the same and we define $(z_i^c - |O_i'|) + z_i^d$ artificial outliers in $(O_{i+1}' \cup \ldots \cup O_k') \cap P_{i+1}$.

Now, for both cases, we apply charging rule I to any points in the remaining modified balls $O_{i+1}' \cup \ldots \cup O_k'$ that are inside $\mathcal{B}(c_i, 4r)$. These points are then marked as covered. See Figure 3 for an example. Note that these points will not be in $P_{i+1} = P_i \backslash \mathcal{B}(c_i, 4r)$. We assumed that $O_1 \cup O_2 \cup \ldots \cup O_k$ covers all but at most $z$ points. After the charging has taken place and the modified clustering $O_1' \cup O_2' \cup \ldots \cup O_k'$ has been constructed, we have that $|O_1' \cup O_2' \cup \ldots \cup O_k'| = n - z - \sum_{j \text{ in case } 1} (z_j^c + z_j^d) - \sum_{j \text{ in case } 2} (z_j^c - |O_j'| + z_j^d)$.

Note that $|O_j'|$ refers to the number of points covered by the modified ball $O_j'$ at the time of iteration $j$. Then, by the way we have charged $O_1' \cup \ldots \cup O_k'$ to our solution $\mathcal{B}(c_1, 4r) \cup \ldots \cup \mathcal{B}(c_k, 4r)$, we obtain $|\mathcal{B}(c_1, 4r) \cup \ldots \cup \mathcal{B}(c_k, 4r)| \geq |O_1' \cup O_2' \cup \ldots \cup O_k'| - \frac{\epsilon z}{\alpha} + \sum_{j \text{ in case } 1} (z_j^c + z_j^d)$
$+ \sum_{j \text{ in case } 2} (z_j^c - |O_j'| + z_j^d) = n - (1 + \frac{\epsilon}{\alpha}) z$. This concludes the proof of Part ① of the lemma. It follows easily that when $r_{\text{OPT}} \leq r < (1 + \epsilon) r_{\text{OPT}}$, Part ② also holds, which completes the proof of this lemma. $\square$

**Lemma 3.8.** *Let $i$ be an iteration of the charging argument above such that we are in case 2. This means that $\mathcal{B}(c_1, 2r) \cup \ldots \cup \mathcal{B}(c_i, 2r)$ does not intersect any of the remaining modified balls. Then, there must be a remaining modified ball covering $\leq \frac{n_i - z}{k - i + 1}$ points.*

For the proof of Lemma 3.8, see Supplementary Section C.

**Lemma 3.9** (Coverage of $B(c_i, 2r)$)**.** *When we are in case 2 of the charging argument for some iteration $i$, we must have that $|\mathcal{B}(c_i, 2r)| \geq \frac{n_i - z}{k-i+1} - \frac{\epsilon z}{\alpha k}$.*

For the proof of Lemma 3.9, see Supplementary Section C.

## 3.3 Duration of dense clusters

After running Procedure 5 as a subroutine, we know that each cluster $C_i$ covers at least $\phi$ points. More specifically, for each level $i$, if $z+1 \leq \frac{n_i - z}{4(k-i+1)}$, then $|\mathcal{B}(c_i, 2r)| \geq \frac{n_i - z}{2(k-i+1)}$ and if $z+1 > \frac{n_i - z}{4(k-i+1)}$, then $|\mathcal{B}(c_i, 2r)| \geq \frac{n_i - z}{k-i+1} - \frac{\epsilon z}{\beta k}$. The following two lemmas show that when $C_i$ satisfies one of these two constraints, it will be dense for a significant number of arbitrary operations. This will lead to the amortized update time being independent of $n$. The remaining lemmas are split into two cases, corresponding to the two cases in Procedure 5.

**Lemma 3.10** (Duration of dense cluster for $z$ is small)**.** *Assume that we are currently at time $t$. Let us consider a level $i$ in which $z + 1 \leq \frac{n_i^t - z}{4(k-i+1)}$. Let $p = \arg\max_{p' \in S_i} |\mathcal{B}_{P_i}(p', 2r)|$, and $\mathcal{B}_{\max} = \mathcal{B}_{P_i}(p, 2r)$. Assume that $\frac{n_i^t - z}{2(k-i+1)} \leq |\mathcal{B}_{max}|$. Then, we can add $\mathcal{B}_{P_i}(p, 4r)$ as a cluster in our solution, and this cluster will be dense until time $t' = t + t^*$, with $t^* \geq \frac{n_i^t - z}{4(k-i+1)}$.*

For the proof of Lemma 3.10, see Supplementary Section D.

**Lemma 3.11** (Duration of dense cluster for $z$ is large)**.** *Assume that we are currently at time $t$. Let $z + 1 > \frac{n_i^t - z}{4(k-i+1)}$ for some level $i$. Let $p = \arg\max_{p' \in S_i} |\mathcal{B}_{P_i}(p', 2r)|$, and $\mathcal{B}_{\max} = \mathcal{B}_{P_i}(p, 2r)$. Assume that $\frac{n_i^t - z}{k-i+1} - \frac{\epsilon z}{\beta k} \leq |\mathcal{B}_{max}|$. Then, we can add $\mathcal{B}_{P_i}(p, 4r)$ as a cluster in our solution, and this cluster will be dense until time $t' = t + t^*$, with $t^* = \Omega(\frac{\epsilon z}{k})$.*

For the proof of Lemma 3.11, see Supplementary Section D.

## 3.4 Small radius guesses

It has not yet been considered what will happen if Procedure 5 fails on some level $i$. That is, when there exists no $p \in S_i$ such that $\mathcal{B}_{P_i}(p, 2r)$ covers sufficiently many points. Lemmas E.1 and E.2 show that if Procedure 5 fails at some level, then with high probability the guess of the optimal radius for that specific instance is too small ($r < r_{\mathrm{OPT}}$). In this case, we postpone the construction of that level to a time $t + t^*$, referred to as $t'_r$ in the algorithm.

## 3.5 Computing update time

Finally, we prove the update time of our algorithm. Lemma 3.4, Lemma 3.5 and Lemma 3.6 prove that the algorithms maintain the invariants with high probability. Together with Lemma 3.7 and Lemma F.1 this implies Theorem 1.1.

## 3.6 Robustness to adversarial inputs

Our dynamic algorithm is robust against an adaptive adversary. Specifically, we do not assume that the adversary has predetermined the sequence of updates in advance, as with an oblivious adversary. Instead, the adversary can query the insertion and deletion updates in an online manner, with knowledge of our solution.

We only use randomization to generate the sample set $S_i$ in Procedure 5, which is then used to construct the levels. This randomness affects only the probability of failure in this procedure. After the insertion or deletion updates, we do not need to reconstruct a level until it no longer satisfies the invariants. The guarantees we provide for how long a level can remain valid are all in the worst case; therefore, they hold even when an adaptive adversary chooses the insertion and deletion operations online.

### 3.7 Acknowledgements

The authors would like to thank the anonymous reviewers for their insightful comments. Annika Hennes' and Melanie Schmidt's research was funded by the Deutsche Forschungsgemeinschaft (DFG, German Research Foundation) – project number 456558332.

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

# A Missing Pseudocode

In this section, we provide the missing pseudocodes. We start by Procedure INITIALIZATION, which is called to start the algorithm. This procedure defines the set $\mathcal{R}$ and then creates a counter for time $t$, and the elements of our data structure $\mathcal{F}_r$, $\mathcal{L}_r$, and $\mathcal{Z}_r$ that we discussed before. It also creates the variable $t'_r$, which is used when the radius $r$ is smaller than $r_{\text{OPT}}$. If $r < r_{\text{OPT}}$, we may fail to find the $p^*$ that we desired for a level $i$ in Procedure OFFLINECLUSTER, and then have to stop the level construction. In this case, we set $t'_r$ to $t + t^*$ to postpone the construction of that level to time $t + t^*$. See Lemmas E.1 and E.2 for the choice of $t^*$. If $t'_r$ is set to $-1$, it means there is currently no construction to be postponed. We assume all the parameters and variables defined in Procedure INITIALIZATION are global, and the other procedures have access to them.

Then, we state Procedure UPDATE to handle the insertion or deletion of a point $p$. Depending on whether the query is an insertion or a deletion, it calls INSERT$(p, r)$ or DELETE$(p, r)$ for different values of $r$. Then it updates the counter for time. Next, if there is any level construction that was postponed to the current time $t$, it handles it.

---

**Procedure 1** INITIALIZATION$(k, z, \epsilon, d_{\min}, d_{\max})$

---
1: $t \leftarrow 0$
2: $\mathcal{R} := \{(1 + \epsilon)^i : d_{\min} \leq (1 + \epsilon)^i \leq (1 + \epsilon)d_{\max}, i \in \mathbb{N}\}$
3: **for all** $r \in \mathcal{R}$ **do**
4: $\quad$ $\mathcal{F}_r \leftarrow \emptyset$, $\mathcal{L}_r \leftarrow \emptyset$, $\mathcal{Z}_r \leftarrow \emptyset$
5: $\quad$ $t'_r \leftarrow -1$
6: **end for**

---

**Procedure 2** UPDATE$(p)$

---
1: **for all** $r \in \mathcal{R}$ **do**
2: $\quad$ **if** UPDATE$(p)$ is an insertion **then**
3: $\quad\quad$ INSERT$(p, r)$
4: $\quad$ **else if** UPDATE$(p)$ is a deletion **then**
5: $\quad\quad$ DELETE$(p, r)$
6: $\quad$ **end if**
7: **end for**
8: $t \leftarrow t + 1$
9: **for all** $r \in \mathcal{R}$ **do**
10: $\quad$ **if** $t'_r = t$ **then**
11: $\quad\quad$ $\lambda_r \leftarrow |\mathcal{F}_r|$
12: $\quad\quad$ OFFLINECLUSTER$(\mathcal{Z}_r, \lambda_r + 1, r)$
13: $\quad$ **end if**
14: **end for**

---

**Procedure 3** INSERT$(p, r)$

---
1: $\lambda \leftarrow |\mathcal{F}_r|$, $\{c_1, c_2, ..., c_\lambda\} \leftarrow \mathcal{F}_r$, and $\{C_1, C_2, ..., C_\lambda\} \leftarrow \mathcal{L}_r$
2: **if** $d(p, c_i) \leq 4r$ for some $i \in [\lambda]$ **then**
3: $\quad$ $C_i \leftarrow C_i \cup p$
4: **else**
5: $\quad$ $\mathcal{Z}_r \leftarrow \mathcal{Z}_r \cup p$
6: **end if**
7: For each level $\ell$, $P_\ell \leftarrow (\cup_{\ell \leq j \leq \lambda} C_j) \cup \mathcal{Z}_r$
8: **for** $i = 1$ to $\lambda$ **do**
9: $\quad$ **if** $|C_i| \leq \min\left(z + 1, \frac{|P_i| - z}{k - i + 1} - \frac{\epsilon z}{\alpha k}\right)$ **then**
10: $\quad\quad$ OFFLINECLUSTER$(P_i, i, r)$
11: $\quad\quad$ break
12: $\quad$ **end if**
13: **end for**

---

**Procedure 4** DELETE$(p, r)$

---

1: $\lambda \leftarrow |\mathcal{F}_r|$
2: **if** $p \in \mathcal{Z}_r$ **then**
3:     remove $p$ from $\mathcal{Z}_r$
4: **else**
5:     Let $C_i$ be the cluster containing $p$, where $i \in [\lambda]$
6:     $P_i \leftarrow (\cup_{i \leq j \leq \lambda} C_j) \cup \mathcal{Z}_r$
7:     **if** $\mathcal{B}(c_i, 2r)$ covers $> \min\left(z + 1, \frac{|P_i| - z}{k - i + 1} - \frac{\epsilon z}{\alpha k}\right)$ from $P_i$ **then**
8:         remove $p$ from $C_i$
9:     **else**
10:         OFFLINECLUSTER$(P_i, i, r)$
11:     **end if**
12: **end if**

---

**Procedure 5** OFFLINECLUSTER$(P_i, i, r)$

---

1: $t'_r \leftarrow -1$
2: **while** $i \leq k$ and $P_i \neq \emptyset$ **do**
3:     $n_i \leftarrow |P_i|$
4:     Pick a uniform sample $S_i$ of $\psi \epsilon^{-1} k^2 \log k$ points from $P_i$
5:     **if** $z + 1 \leq \frac{n_i - z}{4(k - i + 1)}$ **then**
6:         Find a point $p^* \in S_i$ such that $|\mathcal{B}_{P_i}(p^*, 2r)| \geq \frac{n_i - z}{2(k - i + 1)}$
7:         **if** there does not exist such a $p^*$ **then**
8:             $t'_r \leftarrow t + \frac{n_i - z}{2(k - i + 2)}$                 // See Lemma E.1
9:             break
10:         **end if**
11:     **else if** $z + 1 > \frac{n_i - z}{4(k - i + 1)}$ **then**
12:         Find a point $p^* \in S_i$ such that $|\mathcal{B}_{P_i}(p^*, 2r)| \geq \frac{n_i - z}{k - i + 1} - \frac{\epsilon z}{\beta k}$
13:         **if** there does not exist such a $p^*$ **then**
14:             $t'_r \leftarrow t + \frac{\epsilon z}{2\beta k}$                    // See Lemma E.2
15:             break
16:         **end if**
17:     **end if**
18:     $c_i \leftarrow p^*$, $C_i \leftarrow \mathcal{B}(p^*, 4r)$, and $P_{i+1} \leftarrow P_i \backslash C_i$
19:     $i \leftarrow i + 1$
20: **end while**
21: $\mathcal{F}_r \leftarrow \{c_1, c_2, ..., c_{i-1}\}$, $\mathcal{L}_r \leftarrow \{C_1, C_2, ..., C_{i-1}\}$, $\mathcal{Z}_r \leftarrow P_i$

---

# B   Missing proofs Section 3.1

**Lemma 3.4** (Procedure 3 maintains invariants). *Assume that at time $t$, we have point set $P^t$, data structure $\mathcal{D}_r = (\mathcal{F}_r, \mathcal{L}_r, \mathcal{Z}_r)$. We assume that the level and dense invariants hold at time $t$ and $r \geq r_{\text{OPT}}^{t+1}$. At the start of time $t + 1$, we insert point $p$ using Procedure 3. After the insertion, the level and dense invariants still hold with probability 1 if Procedure 5 was not called and with the probability of at least $1 - \frac{2(k - i + 1)}{e^{\Psi \log k}}$, where $\Psi \geq 1$ if Procedure 5 was not called.*

*Proof.* If we enter the case in line 2, the new point $p$ will be added to an existing cluster $C_i$. Then, we have that $p \in P_j$ for all $j \leq i$ and $p \notin P_j$ for all $j > i$. If we enter the case in line 4, the new point $p$ is added as an outlier. This is the highest level, so we have that $p \in P_i$ for all $i$. The level invariant is maintained by definition in these cases. If we enter the case in line 9 and recluster levels $i, \ldots, k$, we recluster the points in $P_i$ as defined in line 7, and hence, none of the levels $j < i$ are affected. Then by Lemma 3.6, the level invariant is maintained for all levels. The dense invariant is maintained because of the check we do in line 9 for all levels. Furthermore, we choose the lowest level $i$ where the dense invariant does not hold and recluster from this level upwards. Hence, the

dense invariant will hold for all levels $j < i$. We call Procedure 5 on $P_i$, so by Lemma 3.6, the dense invariant is maintained for levels $j \geq i$ with probability $1 - \frac{2(k-i+1)}{e^{\Psi \log k}}$, with $\Psi \geq 1$. □

**Lemma 3.5** (Procedure 4 maintains invariants)**.** *Assume that at time $t$, we have point set $P^t$, instance $\mathcal{D}_r = (\mathcal{F}_r, \mathcal{L}_r, \mathcal{Z}_r)$, parameters $k, z \in \mathbb{N}$ and $\epsilon > 0$. We assume that the level and dense invariants hold at time $t$ and $r \geq r_{\text{OPT}}^{t+1}$. At the start of time $t + 1$, we delete an arbitrary point $p$ using Procedure 4. After the deletion, the level and dense invariants hold with probability 1 if Procedure 5 was not called, and with probability $1 - \frac{2(k-i+1)}{e^{\Psi \log k}}$ with $\Psi \geq 1$ if Procedure 5 was called.*

*Proof.* If we enter the case in line 2, where $p$ is an outlier, it follows easily that the level and dense invariants are maintained. In the second case, starting in line 4, $p$ is in some cluster $C_i$, either as a center or as another point. If $C_i$ still covers sufficiently many points after the deletion of $p$ as described in line 7, it follows easily that the level invariant is maintained. For the dense invariant, note that for all levels $j < i$, $n_j$ and with this $\frac{n_j - z}{k - j + 1} - \frac{\epsilon z}{\alpha k}$ can only decrease. Furthermore, for all levels $j > i$, $n_j$ remains unchanged. This observation, combined with the fact that for any $i \neq j$, $|C_j|$ is unchanged, means that the dense invariant still holds for all levels.

If $C_i$ is no longer dense after deleting $p$, and we enter the case in line 9, the clusters in levels $j < i$ remain unchanged. The level invariant will be maintained since $P_j$ is updated to $P_j \backslash \{p\}$ for all $j < i$. The dense invariant is maintained in levels $j < i$ by the same reasoning as the previous case. On the remaining points, Procedure 5 is called in line 10. Using Lemma 3.6, the level and dense invariants are maintained for the remaining levels with probability at least $1 - \frac{2(k-i+1)}{e^{\Psi \log k}}$, where $\Psi \geq 1$. Crucial for maintaining the level invariant is line 6. This line ensures that we do not consider the points in levels $j < i$ when constructing level $i$ and any higher levels. □

**Lemma 3.6** (Procedure 5 maintains invariants with high probability)**.** *Suppose the level and dense invariants hold for all levels $j < i$ and we call Procedure 5 on $P_i$ as the result of an insertion or deletion. Let $\lambda \leq k$ be a random variable representing the number of levels we have after completing Procedure 5. If $r \geq r_{\text{OPT}}$, Procedure 5 maintains the level and dense invariants for all levels $j$ with $i \leq j \leq \lambda$ with probability at least $1 - \frac{2(k-i+1)}{e^{\Psi \log k}}$, with $\Psi \geq 1$.*

*Proof.* By the while-loop structure in combination with line 18, the level invariant is maintained for all levels $j \geq i$. For each newly constructed level $j \geq i$, there are two cases within the while-loop.

For the first case where $z + 1 \leq \frac{n_j - z}{4(k-j+1)}$, we find a ball $\mathcal{B}_{P_j}(p^*, 2r)$ that covers $\geq \frac{n_j - z}{2(k-j+1)}$ points with probability at least $1 - \frac{2}{e^{\Psi \log k}}$ if $r \geq r_{\text{OPT}}$. For the proof of this, we refer to Lemma E.1.

Then, the dense invariant holds for any such level since

$$\frac{n_j - z}{2(k - j + 1)} \geq \frac{n_j - z}{4(k - j + 1)} \geq z + 1 \geq \min\left(z + 1, \frac{n_j - z}{k - j + 1} - \frac{\epsilon z}{\alpha k}\right). \tag{1}$$

In the second case where $z + 1 > \frac{n_j - z}{4(k-j+1)}$, we find a ball $\mathcal{B}_{P_j}(p^*, 2r)$ that covers $\geq \frac{n_j - z}{k-j+1} - \frac{\epsilon z}{\beta k}$ points with probability $1 - \frac{2}{e^{\Psi \log k}}$ if $r \geq r_{\text{OPT}}$. For the proof of this, we refer to Lemma E.2. Since $\beta > \alpha$ (see Section 2.4), the dense invariant also holds for every level in this case. Then, the probability that the dense invariant holds for all levels $j$ with $i \leq j \leq \lambda$ is at least $1 - \frac{2(k-i+1)}{e^{\Psi \log k}}$. This is because $\lambda - i + 1 \leq k - i + 1$ is the number of new levels we constructed during Procedure 5. □

## C  Missing proofs Section 3.2

**Lemma 3.8.** *Let $i$ be an iteration of the charging argument above such that we are in case 2. This means that $\mathcal{B}(c_1, 2r) \cup \ldots \cup \mathcal{B}(c_i, 2r)$ does not intersect any of the remaining modified balls. Then, there must be a remaining modified ball covering $\leq \frac{n_i - z}{k - i + 1}$ points.*

*Proof.* We prove this using strong induction on iterations in case 2. First, let us consider the base case. In the base case, we are in iteration $i$ which is the first iteration to be in case 2 of the charging argument. There have been $i - 1$ iterations before $i$ which were charged according to case 1. For any

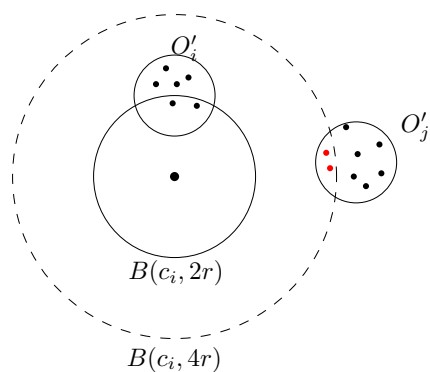

Figure 3: An example to illustrate the extra charging we do at the end of both cases 1 and 2. We are in iteration $i$, and the ball $\mathcal{B}(c_i, 4r)$ covers two points from ball $O'_j$, with $j > i$. The two points, shown in red, will be charged to themselves as in charging *rule I* and marked as covered.

$j < i$, define $x_j$ to be the number of artificial outliers covered by $\mathcal{B}(c_j, 4r)$. Then, for the remaining modified balls we have that:

$$|O'_i \cup \ldots \cup O'_k| = n_i - \left( z - \sum_{j=1}^{i-1}(z_j^c - x_j) \right) - \sum_{j=1}^{i-1}(z_j^c - x_j) = n_i - z \qquad (2)$$

This is because we assume that there are exactly $z$ points outside our optimal solution $O_1 \cup \ldots \cup O_k$. These $z$ points are also outside $O'_1 \cup \ldots \cup O'_k$ since for $1 \le i \le k$, $O'_i \subseteq O_i$. Of those $z$ points, $(z - \sum_{j=1}^{i-1}(z_j^c - x_j))$ have not yet been seen in an iteration. Hence, these points are definitely not covered by $O'_i \cup \ldots \cup O'_k$. Furthermore, we know that $\sum_{j=1}^{i-1}(z_j^c - x_j)$ artificial outliers have been defined in $O'_i \cup \ldots \cup O'_k$ in total. This is because all iterations $j < i$ were in case 1, and any artificial outliers present in $O'_j$ will be propagated to $(O'_{j+1} \cup \ldots \cup O'_k) \cap P_{j+1}$, as these are included in $z_j^c$. We subtract $x_j$ because these artificial outliers have already been counted by another $z_j^c$, and should not be counted again when computing the total amount of artificial outliers. Note that $z_j^d = 0$ for all iterations $j < i$ as there were no previous iterations before $j$ in case 2. Since at most $n_i - z$ points are covered by $O'_i \cup \ldots \cup O'_k$, there must be at least one of the modified optimal ball which covers $\le \frac{n_i - z}{k - i + 1}$. This concludes the proof of the base case.

Now, consider the inductive step. We are in an iteration $i$ that is in case 2. There have been $i - 1$ previous iterations, of which an arbitrary number has been charged by case 2. Assume that for any such iteration $j < i$ that was in case 2, we found an optimal ball $O'_j$ that was covering at most $\frac{n_j - z}{k - j + 1}$ points. At the time of iteration $j$, the points of ball $O'_j$ were charged to distinct points in $\mathcal{B}(c_j, 2r)$, but not necessarily covered. In the iterations between $j$ and $i$, however, more points of the ball $O'_j$ may have been covered. Define $O'_{j,\text{unc}}$ as the points of $O'_j$ that are still uncovered at the time of iteration $i$. Now, for the remaining modified balls, we should have that:

$$|O'_i \cup \ldots \cup O'_k| = n_i - \left( z - \sum_{j=1}^{i-1}(z_j^c - x_j) \right) - \sum_{j < i \text{ in case 1}}(z_j^c + z_j^d - x_j)$$

$$- \sum_{j < i \text{ in case 2}}(z_j^c + z_j^d - |O'_j| - x_j) - \sum_{j < i \text{ in case 2}}|O'_{j,\text{unc}}| \qquad (3)$$

Similar to the base case, $(z - \sum_{j=1}^{i-1}(z_j^c - x_j))$ points will definitely be outside $O'_i \cup \ldots \cup O'_k$. In the iterations before $j < i$ in case 1, $(z_j^c + z_j^d)$ points are stored as a credit. We subtract $x_j$ from this since we don't want to recount outliers from previous iterations. In each iteration $j < i$ that was in case 2, $(z_j^c + z_j^d - |O'_j|)$ are stored as a credit. Even though $B(c_j, 2r)$ is disjoint from the remaining modified balls in this case, $B(c_j, 2r)$ can still cover some artificial outliers present in the remaining modified balls. We do not want to recount these for the total amount of outliers in $O'_i \cup \ldots \cup O'_k$, and hence we subtract $x_j$. Lastly, since the modified balls are disjoint, any points of $O'_j$ with $j < i$ and iteration $j$ in case 2 that are still in the point set $n_i$ are not covered by $O'_i \cup \ldots \cup O'_k$. We can rewrite

this as follows:

$$|O_i' \cup \ldots \cup O_k'| = n_i - z - \left( \sum_{j<i} z_j^d + \sum_{j<i \text{ in case 2}} |O_{j,\text{unc}}'| \right) + \sum_{j<i \text{ in case 2}} |O_j'| = n_i - z \quad (4)$$

This is because by the definition of $z_j^d$, $(\sum_{j<i} z_j^d + \sum_{j<i \text{ in case 2}} |O_{j,\text{unc}}'|) = \sum_{j<i \text{ in case 2}} |O_j'|$. Since at most $n_i - z$ points are covered by $O_i' \cup \ldots \cup O_k'$, there must be at least one of the remaining modified balls which covers $\leq \frac{n_i-z}{k-i+1}$. This concludes the proof. $\qquad \square$

**Lemma 3.9** (Coverage of $B(c_i, 2r)$). *When we are in case 2 of the charging argument for some iteration $i$, we must have that $|\mathcal{B}(c_i, 2r)| \geq \frac{n_i-z}{k-i+1} - \frac{\epsilon z}{\alpha k}$.*

*Proof.* We prove the statement by induction on iterations $i \leq k$ in case 2 of the charging argument. For the base case, iteration $i$ is the first iteration in case 2. This means that for all previous iterations $j < i$, $O_j'$ has been charged and covered by charging rule I. Hence, $\mathcal{B}(c_i, 2r)$ cannot intersect any such $O_j'$ since the points in $O_j'$ are not in $P_i$. We know that $\mathcal{B}(c_i, 2r)$ also does not intersect any of the remaining modified balls by our assumption that iteration $i$ is in case 2. Hence, $\mathcal{B}(c_i, 2r)$ does not intersect $O_1' \cup \ldots \cup O_k'$. The ball $\mathcal{B}(c_i, 2r)$ can cover $x \leq \sum_{j=1}^{i-1}(z_j^c - x_j)$ artificial outliers, where $x$ is the total number of artificial outliers present in $O_i' \cup \ldots \cup O_k'$. The ball $\mathcal{B}(c_i, 2r)$ can cover at most $z - \sum_{j=1}^{i-1}(z_j^c - x_j)$ other points, since there are exactly $z$ points outside $O_1 \cup \ldots \cup O_k$, of which $\sum_{j=1}^{i-1}(z_j^c - x_j)$ have been seen in a previous ball $\mathcal{B}(c_j, 4r)$, for $j < i$. Hence, $|\mathcal{B}(c_i, 2r)| \leq z$. So, in order to satisfy the dense invariant, $|\mathcal{B}(c_i, 2r)| \geq \frac{n_i-z}{k-i+1} - \frac{\epsilon z}{\alpha k}$.

Now consider the induction case. We are in an arbitrary iteration $i$ in case 2, and there have been an arbitrary number of iterations $j < i$ in case 2. For iterations $j < i$ in case 2, we assume that $|\mathcal{B}(c_j, 2r)| \geq \frac{n_j-z}{k-j+1} - \frac{\epsilon z}{\alpha k}$. Even though $\mathcal{B}(c_i, 2r)$ is disjoint from the remaining modified balls, it can cover $x$ artificial outliers and $z - \sum_{j=1}^{i-1}(z_j^c - x_j)$ other points. For $x$ we have that:

$$x \leq \sum_{j<i \text{ in case 1}} (z_j^c + z_j^d - x_j) + \sum_{j<i \text{ in case 2}} (z_j^c + z_j^d - |O_j'| - x_j) \quad (5)$$

$\sum_{j<i \text{ in case 1}}(z_j^c + z_j^d - x_j)$ are the total number of artificial outliers from iterations in case 1, and $\sum_{j<i \text{ in case 2}}(z_j^c + z_j^d - |O_j'| - x_j)$ are the total number of artificial outliers from case 2. Then, for the total amount of points in $\mathcal{B}(c_i, 2r)$, we have:

$$|\mathcal{B}(c_i, 2r)| \leq z + \sum_{j<i} z_j^d - \sum_{j<i \text{ in case 2}} |O_j'| \leq z \quad (6)$$

This is because by the definition of $z_j^d$, we have that $\sum_{j<i} z_j^d \leq \sum_{j<i \text{ in case 2}} |O_j'|$. So, in order to satisfy the dense invariant, $|\mathcal{B}(c_i, 2r)| \geq \frac{n_i-z}{k-i+1} - \frac{\epsilon z}{\alpha k}$. This concludes the proof. $\qquad \square$

## D Missing proofs Section 3.3

**Lemma 3.10** (Duration of dense cluster for $z$ is small). *Assume that we are currently at time $t$. Let us consider a level $i$ in which $z + 1 \leq \frac{n_i^t-z}{4(k-i+1)}$. Let $p = arg\max_{p' \in S_i} |\mathcal{B}_{P_i}(p', 2r)|$, and $\mathcal{B}_{\max} = \mathcal{B}_{P_i}(p, 2r)$. Assume that $\frac{n_i^t-z}{2(k-i+1)} \leq |\mathcal{B}_{max}|$. Then, we can add $\mathcal{B}_{P_i}(p, 4r)$ as a cluster in our solution, and this cluster will be dense until time $t' = t + t^*$, with $t^* \geq \frac{n_i^t-z}{4(k-i+1)}$.*

*Proof.* The subscript of $n_i$ will be omitted for simplicity. At time $t$, we know that $\frac{n^t-z}{2(k-i+1)} \leq |\mathcal{B}_{\max}|$. The resulting cluster $\mathcal{B}_{P_i}(p, 4r)$ will be dense as long as $\mathcal{B}_{\max}$ covers more than $\min\left(z + 1, \frac{n^{t'}-z}{k-i+1} - \frac{\epsilon z}{\alpha k}\right)$ points at time $t'$. Hence, the dense invariant will not be broken as long as

$\mathcal{B}_{\max}$ is greater than $z + 1$. Let $t'$ be the time at which $|\mathcal{B}_{\max}| < z + 1$. We can lower bound the time $t^*$ between $t$ and $t'$ using the change in size of $\mathcal{B}_{\max}$. Then, we get the following lower bound:

$$t^* \geq \frac{n^t - z}{2(k - i + 1)} - z \tag{7}$$

Then, since $z + 1 \leq \frac{n^t - z}{4(k - i + 1)}$ by our assumption, we have:

$$t^* > \frac{n^t - z}{2(k - i + 1)} - \frac{n^t - z}{4(k - i + 1)} \geq \frac{n^t - z}{4(k - i + 1)} \tag{8}$$

This concludes the proof. $\square$

**Lemma 3.11** (Duration of dense cluster for $z$ is large). *Assume that we are currently at time $t$. Let $z + 1 > \frac{n_i^t - z}{4(k - i + 1)}$ for some level $i$. Let $p = arg\,max_{p' \in S_i}|\mathcal{B}_{P_i}(p', 2r)|$, and $\mathcal{B}_{\max} = \mathcal{B}_{P_i}(p, 2r)$. Assume that $\frac{n_i^t - z}{k - i + 1} - \frac{\epsilon z}{\beta k} \leq |\mathcal{B}_{max}|$. Then, we can add $\mathcal{B}_{P_i}(p, 4r)$ as a cluster in our solution, and this cluster will be dense until time $t' = t + t^*$, with $t^* = \Omega(\frac{\epsilon z}{k})$.*

*Proof.* The subscript of $n_i$ will be omitted for simplicity. At time $t$, we know that $\frac{n^t - z}{k - i + 1} - \frac{\epsilon z}{\beta k} \leq |\mathcal{B}_{\max}|$. The resulting cluster $\mathcal{B}_{P_i}(p, 4r)$ will be dense as long as $\mathcal{B}_{\max}$ covers more than $\min\left(z + 1, \frac{n^{t'} - z}{k - i + 1} - \frac{\epsilon z}{\alpha k}\right)$ points at time $t'$, with $\alpha$ a fixed constant smaller than $\beta$. Hence, the dense invariant will not be broken as long as $\mathcal{B}_{\max}$ covers at least $\frac{n^{t'} - z}{k - i + 1} - \frac{\epsilon z}{\alpha k}$ points. Let $t'$ be the time at which $|\mathcal{B}_{\max}| < \frac{n^{t'} - z}{k - i + 1} - \frac{\epsilon z}{\alpha k}$. We can lower bound $t^*$ by examining the change in size of $\mathcal{B}_{\max}$. Using that $n^{t'} \leq n^t + t^*$, we have:

$$\begin{aligned} t^* &\geq \left(\frac{n^t - z}{k - i + 1} - \frac{\epsilon z}{\beta k}\right) - \left(\frac{n^{t'} - z}{k - i + 1} - \frac{\epsilon z}{\alpha k}\right) \\ &\geq \frac{n^t - z}{k - i + 1} - \frac{n^t + t^* - z}{k - i + 1} - \frac{\epsilon z}{\beta k} + \frac{\epsilon z}{\alpha k} = \frac{-t^*}{k - i + 1} - \frac{\epsilon z}{\beta k} + \frac{\epsilon z}{\alpha k} \end{aligned} \tag{9}$$

Now, solving for $t^*$ gives:

$$t^* \geq \frac{\epsilon z (k - i + 1)}{k(k - i + 2)}\left(\frac{1}{\alpha} - \frac{1}{\beta}\right) \geq \frac{\epsilon z}{2k}\left(\frac{1}{\alpha} - \frac{1}{\beta}\right) \tag{10}$$

Since $\alpha$ and $\beta$ are two fixed constants such that $\beta > \alpha$, this gives $t^* = \Omega(\frac{\epsilon z}{k})$. This concludes the proof. $\square$

## E  Missing proofs Section 3.4

**Lemma E.1** (Radius guess is small for $z$ is small). *Assume that we are currently at time $t$. Let us consider a level $i$ for which $z + 1 \leq \frac{n_i^t - z}{4(k - i + 1)}$. Let $p = arg\,max_{p' \in S_i}|\mathcal{B}_{P_i}(p', 2r)|$, where $S_i$ is the sample chosen in Algorithm 5, and $\mathcal{B}_{\max} = \mathcal{B}_{P_i}(p, 2r)$. Assume that $|\mathcal{B}_{max}| < \frac{n_i^t - z}{2(k - i + 1)}$. Then, with probability at least $1 - \frac{2}{e^{\Psi \log k}}$, for $\Psi \geq 1$, we have that $r < r_{OPT}$ and until time $t' = t + t^*$ we do not need to consider the instance for $r$, with $t^* \geq \frac{n_i^t - z}{2(k - i + 2)}$.*

*Proof.* Let $O_1, \ldots, O_k$ be the optimal balls at time $t$. Consider the same charging argument used in Lemma 3.7, where we charged one of the optimal balls to $\mathcal{B}_{P_i}(c_i, 4r)$ in each level $i$. Using Equation 4 and the fact that $O_i' \subseteq O_i$ for each of the remaining optimal balls, we know that $|O_i \cup \ldots \cup O_k| \geq n_i^t - z$. Then, there must be at least one of the remaining optimal balls covering $\geq \frac{n_i^t - z}{k - i + 1}$ points. Let $O_i$ be the largest such ball. Using the Chernoff bound, we show that with high probability, one of the sampled points $p \in S_i$ is in $O_i$. Define independent random variables $X_1, \ldots, X_{|O_i|}$, one for each point in $O_i$. Each $X_j$, corresponding to point $j \in O_i$, will be 1 if $j \in S_i$

and 0 otherwise. Define $X = \sum_{j=1}^{|O_i|} X_j$. We know that $\mathbf{E}[X_j] = \frac{|S_i|}{n_i^t}$. Then, using linearity of expectation, we find:

$$\mathbf{E}[X] = \frac{|S_i| \cdot |O_i|}{n_i^t} \geq \frac{|S_i|}{n_i^t} \cdot \frac{n_i^t - z}{k - i + 1} \geq \frac{|S_i| \cdot (n_i^t - z)}{n_i^t \cdot k} \geq \frac{|S_i|}{k}\left(1 - \frac{z}{n_i^t}\right) \tag{11}$$

The condition $z + 1 \leq \frac{n_i^t - z}{4(k-i+1)}$ implies $z \leq \frac{n_i^t}{4(k-i+1)} \leq \frac{n_i^t}{4}$. Then, since $|S_i| = \psi \epsilon^{-1} k^2 \log k$, it follows that $\frac{|S_i|}{k}\left(1 - \frac{z}{n_i^t}\right) \geq \frac{3}{4}\psi\epsilon^{-1} k \log k$. Given that $\psi \geq 6\beta$ and $\beta > \alpha \geq 1$, this is at least $3\epsilon^{-1} k \log k \geq 3\Psi \log k$ for $\Psi \geq 1$.

Now, using the Chernoff bound, we find:

$$\Pr\left[|X - \mathbf{E}[X]| \geq \mathbf{E}[X]\right] \leq 2e^{-\frac{\mathbf{E}[X]}{3}} \leq \frac{1}{e^{\Psi \log k}}, \tag{12}$$

where $\Psi \geq 1$.

Hence, with probability of at least $1 - \frac{2}{e^{\Psi \log k}}$, there will be at least 1 point from $O_i$ in $S_i$. Let us call this point $p$. If $r \geq r_{\text{OPT}}$, $\mathcal{B}_{P_i}(p, 2r)$ would cover all points of $O_i$ since $p \in O_i$ and $O_i$ has radius $\leq r_{\text{OPT}}$. However, since $|\mathcal{B}_{\max}| < \frac{n_i^t - z}{2(k-i+1)}$, we must have that $r < r_{\text{OPT}}$.

Left to prove is that until time $t' = t + t^*$ we do not need this instance of $r$, where $t^* \geq \frac{n_i^t - z}{2(k-i+2)}$. To this end, let $\mathcal{B}_{\max}^t$ be $\mathcal{B}_{\max}$ at time $t$. Consider the situation at time $t$. We know that $|\mathcal{B}_{\max}^t| < \frac{n_i^t - z}{2(k-i+1)}$.

Let $t'$ be some time after $t$ such that $|\mathcal{B}_{\max}^{t'}| \geq \frac{n_i^{t'} - z}{k-i+1}$. Hence, the instance for $r$ becomes valid at time $t'$. We want to derive a lower bound for $t^* = t' - t$ to complete the proof. By examining the change in the size of $\mathcal{B}_{\max}$, we derive the following lower bound:

$$t^* \geq \frac{n_i^{t'} - z}{k - i + 1} - \frac{n_i^t - z}{2(k - i + 1)} \tag{13}$$

Using $n_i^{t'} \geq n_i^t - t^*$ and subsequently solving for $t^*$ we find that:

$$t^* \geq \frac{n_i^t - z}{2(k - i + 2)} \tag{14}$$

This completes the proof. $\square$

**Lemma E.2** (Radius guess is small for $z$ is large). *Assume that we are currently at time $t$. Let us consider a level $i$ in which $z + 1 > \frac{n_i^t - z}{4(k-i+1)}$. Let $p = \arg\max_{p' \in S_i} |\mathcal{B}_{P_i}(p', 2r)|$, where $S_i$ is the sample chosen in Algorithm 5, and $\mathcal{B}_{\max} = \mathcal{B}_{P_i}(p, 2r)$. Assume that $|\mathcal{B}_{max}| < \frac{n_i^t - z}{k-i+1} - \frac{\epsilon z}{\beta k}$. Then, with probability at least $1 - \frac{2}{e^{\Psi \log k}}$, for $\Psi \geq 1$, we have that $r < r_{\text{OPT}}$ and until time $t' = t + t^*$ we do not need to consider the instance for $r$, with $t^* \geq \frac{\epsilon z}{2\beta k}$.*

*Proof.* Let $O_1, \ldots, O_k$ be the optimal balls at time $t$. As in Lemma E.1, let $O_i$ be the largest remaining optimal ball covering $\geq \frac{n_i^t - z}{k-i+1}$ points. Using the Chernoff bound, we show that with high probability, one of the sampled points $p \in S_i$ is in $O_i$. Define independent random variables $X_1, \ldots, X_{|O_i|}$, one for each point in $O_i$. Each $X_j$, corresponding to point $j \in O_i$, will be 1 if $j \in S_i$ and 0 otherwise. Define $X = \sum_{j=1}^{|O_i|} X_j$. We know that $\mathbf{E}[X_j] = \frac{|S_i|}{n_i^t}$. Then, using linearity of expectation, we find

$$\mathbf{E}[X] = \frac{|S_i| \cdot |O_i|}{n_i^t} \geq \frac{|S_i|}{n_i^t} \cdot \frac{n_i^t - z}{k - i + 1}. \tag{15}$$

Without loss of generality, we can assume $\frac{n_i^t - z}{k-i+1} - \frac{\epsilon z}{\beta k} \geq 1$, otherwise there would not need to be any points in the ball $\mathcal{B}_{P_i}(p, 2r)$ to satisfy the dense invariant, and level $i$ would be trivial. Using this, Equation (15) can be simplified as follows:

$$\mathbf{E}[X] \geq \frac{|S_i|}{n_i^t} \cdot \frac{n_i^t - z}{k - i + 1} \geq \frac{|S_i|}{n_i^t} \cdot \frac{\epsilon z}{\beta k} \tag{16}$$

We make a case distinction. Either $z \geq \frac{n_i^t}{2}$. In this case, Equation (15) further simplifies to

$$\mathbf{E}[X] \geq \frac{\epsilon |S_i|}{2\beta k}. \tag{17}$$

Recall that $|S_i| = \psi \epsilon^{-1} k^2 \log k$. Given that $\psi \geq 6\beta$ and $\beta > \alpha \geq 1$, we have

$$\mathbf{E}[X] \geq \frac{1}{2\beta} \psi k \log k \geq 3k \log k \geq 3\Psi \log k \tag{18}$$

for some $\Psi \geq 1$.

In the other case $z < \frac{n_i^t}{2}$, Equation (15) simplifies to

$$\mathbf{E}[X] \geq \frac{|S_i|}{n_i^t} \cdot \frac{n_i^t / 2}{k - i + 1} \geq \frac{|S_i|}{2k} \geq \frac{1}{2} \psi \epsilon^{-1} k \log k \geq 3\Psi \log k \tag{19}$$

for some $\Psi \geq 1$.

In both cases, using the Chernoff bound, we find that with probability at least $1 - \frac{2}{e^{\Psi \log k}}$, there will be at least one point $p$ from $O_i$ in $X_i$. If $r \geq r_{\mathrm{OPT}}$, $\mathcal{B}_{P_i}(p, 2r)$ would cover all points of $O_i$ since $p \in O_i$ and $O_i$ has radius $\leq r_{\mathrm{OPT}}$. However, since $|\mathcal{B}_{\max}| < \frac{n_i^t - z}{k - i + 1} - \frac{\epsilon z}{\beta k} < \frac{n_i^t - z}{k - i + 1} \leq |O_i|$, we must have that $r < r_{\mathrm{OPT}}$.

Left to prove is that until time $t' = t + t^*$, with $t^* = \Omega(\frac{\epsilon z}{k})$, we do not need this instance of $r$. To this end, let $\mathcal{B}_{\max}^t$ be $\mathcal{B}_{\max}$ at time $t$. Consider the situation at time $t$. We know that $|\mathcal{B}_{\max}^t| < \frac{n_i^t - z}{k - i + 1} - \frac{\epsilon z}{\beta k}$.

Let $t'$ be some time after $t$ such that $|\mathcal{B}_{\max}^{t'}| \geq \frac{n_i^{t'} - z}{k - i + 1}$. The cluster $\mathcal{B}_{P_i}(p, 2r)$ in level $i$ is now covering sufficiently many points. We want to derive a lower bound for $t^* = t' - t$ to complete the proof. By examining the change in the size of $\mathcal{B}_{\max}$ and using that $n_i^{t'} \geq n_i^t - t^*$, we derive

$$t^* \geq \frac{n_i^{t'} - z}{k - i + 1} - \left( \frac{n_i^t - z}{k - i + 1} - \frac{\epsilon z}{\beta k} \right) \tag{20}$$

$$= \frac{n_i^t - t^* - z}{k - i + 1} - \frac{n_i^t - z}{k - i + 1} + \frac{\epsilon z}{\beta k} \tag{21}$$

$$= \frac{-t^*}{k - i + 1} + \frac{\epsilon z}{\beta k} \tag{22}$$

as a lower bound for $t^*$. Solving for $t^*$ gives

$$t^* \geq \frac{\epsilon z (k - i + 1)}{\beta k (k - i + 2)} \geq \frac{\epsilon z}{2\beta k}. \tag{23}$$

Since $\beta$ is a constant, this is in $\Omega(\frac{\epsilon z}{k})$. This concludes the proof. □

## F  Missing proofs from Section 3.5

**Lemma F.1.** *The amortized update time of our dynamic algorithm is $\mathcal{O}(\epsilon^{-3} k^6 \log(k) \log(\Delta))$.*

*Proof.* Let us fix an arbitrary time $t$ and assume that at time $t$, there are $\lambda \leq k$ clusters. Let time $t' = t + t^*$ be the time at which we need to invoke Procedure 5 on an arbitrary level $i \leq k$, due to the dense invariant being violated. We have two cases:

❶ $z + 1 \leq \frac{n_i^t - z}{4(k - i + 1)}$: For this case, Lemma 3.10 shows the following: If we find a cluster such that $|\mathcal{B}_{P_i}(c_i, 2r)| \geq \frac{n_i^t - z}{2(k - i + 1)}$ at time $t$, then this cluster will remain dense for $t^* \geq \frac{n_i^t - z}{4(k - i + 1)}$ update operations (either insert or delete).

❷ $z + 1 > \frac{n_i^t - z}{4(k - i + 1)}$: For this case, Lemma 3.11 proves that if we find a cluster such that $|\mathcal{B}_{P_i}(c_i, 2r)| \geq \frac{n_i^t - z}{k - i + 1} - \frac{\epsilon z}{\beta k}$ at time $t$, then this cluster will remain dense for the next $t^* \geq \frac{\epsilon z}{2k}(\frac{1}{\alpha} - \frac{1}{\beta})$ time steps. Asymptotically, $t^* = \Omega(\frac{\epsilon z}{k})$. By substituting the lower bound of $z$ in this formula, we obtain $t^* = \Omega(\frac{\epsilon n_i^t}{k^2})$.

Observe that in both cases, we have a worst-case guarantee for the number of time steps during which the cluster $\mathcal{B}_{P_i}(c_i, 2r)$ remains dense. This, in turn, allows us to achieve a worst-case amortized update time, rather than the weaker notion of expected amortized update time. We analyze the update time of each case separately.

Let us start with the first case which is $z + 1 \leq \frac{n_i^t - z}{4(k-i+1)}$ and suppose that we find a cluster such that $|\mathcal{B}_{P_i}(c_i, 2r)| \geq \frac{n_i^t - z}{2(k-i+1)}$ at time $t$. We know that $t^* \geq \frac{n_i^t - z}{4(k-i+1)}$. The cost of reclustering levels $i, \ldots, k$ according to Lemma 3.3 is $\mathcal{O}(n_i^{t'} \epsilon^{-1} \cdot k^3 \log k)$, with $n_i^{t'} \leq n_i^t + t^*$. Then, the amortized update time of an arbitrary update operation is $\mathcal{O}(\frac{1}{t^*}(n_i^t + t^*)\epsilon^{-1}k^3 \log k) = \mathcal{O}(\frac{n_i^t}{t^*}\epsilon^{-1}k^3 \log k)$. Since $z \leq \frac{n_i^t}{4(k-i+1)}$, we obtain $\frac{n_i^t}{t^*} \leq n_i^t \cdot \frac{4(k-i+1)}{n_i^t - z} \leq 4k \cdot \frac{n_i^t}{n_i^t - z} \leq 4k \cdot \frac{n_i^t}{n_i^t(1 - \frac{1}{4(k-i+1)})} = 4k \cdot \frac{4(k-i+1)}{4(k-i+1)-1} = \mathcal{O}(k)$ . Thus, $\mathcal{O}(\frac{n_i^t}{t^*}\epsilon^{-1}k^3 \log k) = \mathcal{O}(\epsilon^{-1}k^4 \log k)$.

Now, we consider the second case. Using a similar analysis as in the first case, we have $t^* = \Omega(\frac{\epsilon n_i^t}{k^2})$. Then, the amortized update time of an arbitrary update operation is $\mathcal{O}(\frac{n_i^t}{t^*}\epsilon^{-1}k^3 \log k) = \mathcal{O}(\epsilon^{-2}k^5 \log k)$.

After reclustering levels $i, \ldots, k$, there is no longer a lower bound for $t^*$ for levels $j < i$. Thus, it could happen that such a level $j$ needs to be reclustered soon after time $t'$. Since $j < k$, this leads to an extra factor $k$ such that the final amortized cost is $\mathcal{O}(\epsilon^{-2}k^6 \log k)$.

Finally, we need to consider the case that offline clustering fails for a level $i$. That is, we fail to find a ball covering sufficiently many points in Procedure 5. If offline clustering fails on some level $i$ in case 1, meaning we fail to find a ball such that $|\mathcal{B}_{P_i}(c_i, 2r)| \geq \frac{n_i^t - z}{2(k-i+1)}$, then Lemma E.1 proves that with probability $1 - \frac{1}{e^{\Omega(\log k)}}$, the guess $r$ is small (i.e., $r < r_{\mathrm{OPT}}$) and indeed, remains small and we do not need to consider this guess until a time $t' = t + t^*$ where $t^* \geq \frac{n_i^t - z}{2(k-i+2)}$.

At time $t'$, we recluster levels $i, \ldots, k$. Then, since we are in the first case, we have $z \leq \frac{n_i^t}{4(k-i+1)}$. Thus, the amortized update time of an arbitrary update operation in this case is $\mathcal{O}(\frac{1}{t^*}(n_i^t + t^*)\epsilon^{-1}k^3 \log k) = \mathcal{O}(\frac{n_i^t}{t^*}\epsilon^{-1}k^3 \log k) = \mathcal{O}(\epsilon^{-1}k^4 \log k)$ using Lemma 3.3.

Next, we consider the situation when the offline clustering fails in case 2. Specifically, this occurs when we fail to find a ball such that $|\mathcal{B}_{P_i}(c_i, 2r)| \geq \frac{n_i - z}{k-i+1} - \frac{\epsilon z}{\beta k}$. According to Lemma E.2, the guess $r$ is small (i.e., $r < r_{\mathrm{OPT}}$) with probability $1 - \frac{1}{e^{\Omega(\log k)}}$ and indeed, remains small and we do not need to consider this guess until time $t' = t + t^*$ where $t' = \Omega(\frac{\epsilon z}{k})$. This leads to an amortized update time of $\mathcal{O}(\epsilon^{-2}k^5 \log k)$. As mentioned above, the running time needs to be multiplied by an additional factor $k$ to account for the possible need to recluster lower levels.

Finally, we need to consider $\mathcal{O}(\frac{\log \Delta}{\log(1+\epsilon)})$ guesses for the optimal radii. For small $\epsilon$, this is in $\mathcal{O}(\frac{\log \Delta}{\epsilon})$. Thus, we derive the final amortized update time of $\mathcal{O}(\epsilon^{-3}k^6 \log k \log \Delta)$. $\qquad \square$

# G  How to support discrete $k$-center clustering with outliers

**Lemma G.1.** *Let $(M, d)$ be a metric space and $\epsilon > 0$ be an accuracy parameter. The spread ratio $\Delta = \frac{d_{\max}}{d_{\min}}$ of all points ever inserted is assumed to be bounded. There exists a randomized fully dynamic algorithm that maintains a discrete $k$-center solution that allows up to $(1+\epsilon)z$ many outliers on the current set of points. At every point in time $t$, the current clustering with centers $c_1, \ldots, c_\lambda$ is a $(4 + \epsilon)$-approximation to an optimal solution for the $(k, z)$-center problem with high probability and $c_i \in P^t$ for all $i \leq \lambda$. Upon insertion or deletion of a point, the data structure is updated in amortized update time $\mathcal{O}(\epsilon^{-3}k^6 \log(k) \log(\Delta))$.*

*Proof.* Let $r$ and $i$ be fixed, and $c_i$ be the center of cluster $C_i = \mathcal{B}_{P_i}(c_i, 4r)$. As long as $c_i$ is not deleted, we report $c_i$ as the $i$-th center. Note that any feasible solution for the discrete version is also a feasible solution for the non-discrete version. Therefore, the optimal radius for the discrete version is not smaller than the optimal radius for the non-discrete version.

After the deletion of the point $c_i$, we consider two cases: $\min\left(z+1, \frac{n_i-z}{k-i+1} - \frac{\varepsilon z}{\alpha k}\right) \leq 0$ or $\min\left(z+1, \frac{n_i-z}{k-i+1} - \frac{\varepsilon z}{\alpha k}\right) > 0$. If $\min\left(z+1, \frac{n_i-z}{k-i+1} - \frac{\varepsilon z}{\alpha k}\right) \leq 0$, then $n_i < (1+\varepsilon)z$ as $i \geq 1$ and $\alpha > 1$. Hence, we can report all the $n_i$ points as outliers and stop. We next consider the second case. The dense invariant states that $|\mathcal{B}_{P_i}(c_i, 2r)| \geq \min(z+1, \frac{n_i-z}{k-i+1} - \frac{\varepsilon z}{\alpha k})$. Therefore, $|\mathcal{B}_{P_i}(c_i, 2r)| > 0$ holds in the second case, and there exists a point $\hat{p} \in \mathcal{B}_{P_i}(c_i, 2r)$. Then we report an arbitrary point $\hat{p} \in \mathcal{B}_{P_i}(c_i, 2r)$ after the deletion of $c_i$. Note that we do not replace $c_i$ by $\hat{p}$ in our data structure, and $c_i$ does not change in our data structure as long as the $i$-th level is not reconstructed. The point $\hat{p}$ is just the center that we report for the discrete version.

We next prove the 6-approximation guarantee. To do this, we show that $C_i \subseteq \mathcal{B}(\hat{p}, 6r)$. This implies that any feasible solution with radius $4r$ for the non-discrete version can be used to report a feasible solution for the discrete version. Also, see Figure 4 for a visual representation. Let $q \in C_i$. We show that $q \in \mathcal{B}(\hat{p}, 6r)$. Since $q \in C_i = \mathcal{B}(c_i, 4r)$, we have $d(q, c_i) \leq 4r$. Moreover, $d(c_i, \hat{p}) \leq 2r$ since $\hat{p} \in \mathcal{B}(c_i, 2r)$. Then by the triangle inequality, we have

$$d(q, \hat{p}) \leq d(q, c_i) + d(c_i, \hat{p}) \leq 4r + 2r = 6r.$$

To report a solution for the discrete version, it is enough to keep $\mathcal{B}(c_i, 2r) \cap P_i$. Note that our data structure already stores $C_i \cap P_i$, and since $\mathcal{B}(c_i, 2r) \cap P_i \subseteq C_i \cap P_i$, the space complexity and time complexity remain the same. $\qquad\square$

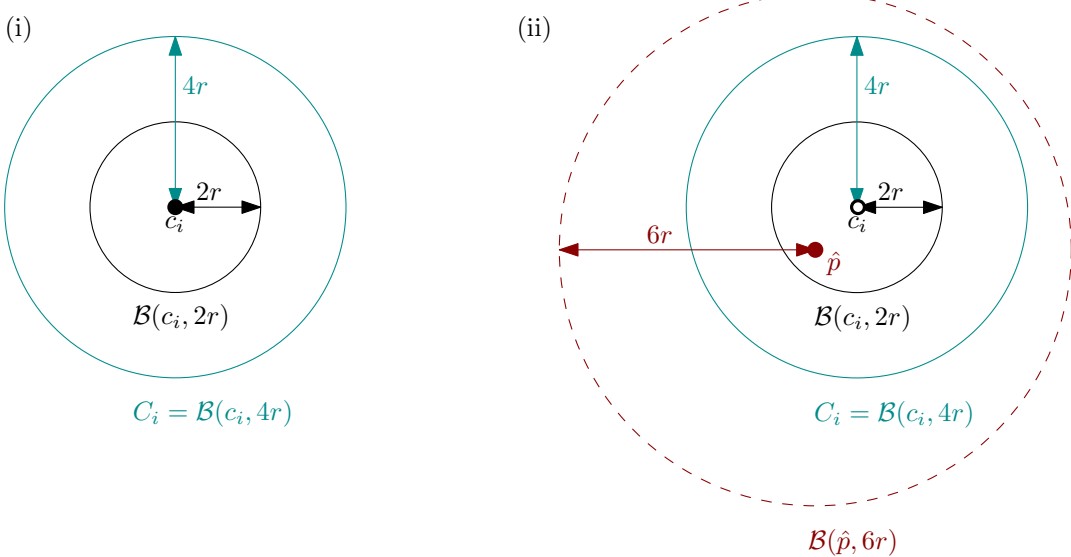

Figure 4: Illustration of the claim's proof. Part (i): before deletion of $c_i$, we report $c_i$ as the center. Part (ii): after deletion of $c_i$, we report an arbitrary point $\hat{p} \in \mathcal{B}(c_i, 2r)$ as the center. Then, $\mathcal{B}(\hat{p}, 6r)$ can cover all the points in the cluster $C_i$.

