# OpenReview forum: "Improved Guarantees for Fully Dynamic $k$-Center Clustering with Outliers in General Metric Spaces"
_NeurIPS.cc/2024/Conference — NeurIPS 2024 poster_

### Official Review · Reviewer_fWP4 · 2024-07-07

**Soundness:** 3
**Presentation:** 3
**Contribution:** 3
**Rating:** 6
**Confidence:** 3

**Summary:**

This paper proposes a simple but effective method that can solve "fully dynamic k center problem with outliers" in general metric space. The fully dynamic setting requires the algorithm to adjust its output efficiently when deletion or insertion operations  occur. This paper uses a "ball cover" strategy to obtain a $(4+\epsilon)$-approximate solution of k center problem with outliers. The radii of balls will be guessed close enough to the optimal k center radius after at most $O(\log \Delta)$ iterations. When a deletion or insertion occurs, the algorithm needs to re-cluster (offline algorithm) if necessary. However, the paper proves that the re-clustering operation will not happen frequently. As a result, even though the time complexity of re-clustering may be as high as $O(n\epsilon^{-1}k^3\log k \log \Delta)$ in worst case, the amortized time can be reduced to $\epsilon^{-2}k^6\log k \log \Delta$ for handling the insertion or deletion of a single point.

**Strengths:**

1. The approximate ratio of the proposed algorithm is $4+\epsilon$, far better than previous paper like [5].
2. The algorithm of this paper do not need extra assumption of metric space, *e.g.*, low doubling dimension assumption in [2], which makes the algorithm have strong applicability.
3. The presentation is nice.

**Weaknesses:**

1. Time complexity. The amortized update time $\epsilon^{-2}k^6\log k \log \Delta$ is worse than the method in [5] (if $k$ is large), whose time complexity is $O(\frac{k^2\log \Delta}{\epsilon^2\tau}\log^2\frac{1}{\delta})$​.
2. High update time complexity in the worst case. The proposed algorithm still need  $O(n\epsilon^{-1}k^3\log k \log \Delta)$  time to handle an insertion or deletion of a single point in the worst case.
3. No experimental result. Although this work is mostly theoretical, it does not means that experiment is not necessary (especially for NeurIPS, not a purely theory venue). The previous work [5] provides the experimental results, so I strongly encourage the authors to compare the performance of both running time and cost to [5].

**Questions:**

1. About the time complexity of binary search of $r_{OPT}$, you have mentioned that the set of all possible $r$ is $R = \{(1+\epsilon)^i:d_{min}\le (1+\epsilon)^i\le d_{max}, i\in N\}$. So $|R| = \log_{1+\epsilon}\Delta = \frac{\log \Delta}{\log (1+\epsilon)}$. So if you binary search in the set $R$, it seems like that you only need $O(\log |R|) = O(\log \log \Delta)$ steps. So can your time complexity improved to $O(\epsilon^{-2}k^6\log k \log \log \Delta)$?
2. If I want to delete some point which is also a center of the solution. I think this center point should also be deleted since this point has been removed from this metric space. But it seems that Procedure 2 in the paper do not consider this situation. How do your algorithm handle the deletion of the center points?
3. What is the size of your memory usage?

**Limitations:**

Please see weakness.

---

> ### Author Rebuttal · Authors · 2024-08-07
>
> **Comment:** Time complexity. The amortized update time $\epsilon^{-2}k^6\log k\log \Delta$ is worse than the method in [5] (if $k$ is large), whose time complexity is $O(\frac{k^2\log\Delta}{\epsilon^2\tau}\log^2\frac{1}{\delta})$.
>
> **Comment:** High update time complexity in the worst case. The proposed algorithm still need $O(n\epsilon^{-1}k^3\log k\log\Delta)$ time to handle an insertion or deletion of a single point in the worst case.
>
> **Response:** Besides the improvement in approximation factor with respect to the algorithm in [5], another interesting aspect of our method is that it is **adversarially robust**, meaning it can handle an adversary who observes the computed solution after each update and adjusts the subsequent updates accordingly. In contrast, the dynamic $14$-approximation algorithm developed by Chan et al. is **not** adversarially robust. In particular, if their solution is revealed at any time, the adversary can impose an expensive update that requires time in $\Omega(n)$.
>
> Regarding the question about handling an insertion or deletion of a single point in the worst case with $O(n \epsilon^{-1} k^3 \log k \log \Delta)$ time, we believe this can be managed with some adjustments to our algorithm. Specifically, we propose running two parallel instances of our algorithm. In the first instance, we compute and output the solution, while the second instance divides the large operations of $O(n \epsilon^{-1} k^3 \log k \log \Delta)$ time into smaller chunks of size $\text{poly}(\epsilon^{-1}, k, \log k, \log \Delta)$ and processes each chunk sequentially after an update. When the solution maintained by the first run becomes invalid—potentially after $\Omega(n)$ updates—we switch to the second run, obtain the solution from there, and continue the process. It is important to note that when switching between runs, we obtain a new solution and maintain that updated solution.
>
> **Comment:** No experimental result. [...] I strongly encourage the authors to compare the performance of both running time and cost to [5].
>
> **Response:**  We agree with the reviewer that providing empirical results or practical evaluations would be valuable and interesting work. Our main focus was to improve the $14$-approximation guarantee and determine how much we could refine the approximation factor, leading to the development of the first adversarially robust dynamic $4$-approximation algorithm for the metric $k$-center problem with outliers. We consider conducting experiments and benchmarking our dynamic algorithm against the algorithm by Chan et al. for both real and synthetic scenarios as future work.
>
> **Question:** About the time complexity of binary search of $r_{OPT}$, you have mentioned that the set of all possible $r$ is $R=\{(1+\epsilon)^i\colon d_{\min} \leq (1+\epsilon)^i \leq d_{\max}, i\in N\}$. So $|R| = \log_{1+\epsilon}\Delta = \frac{\log\Delta}{\log(1+\epsilon)}$. So if you binary search the set $R$, it seems like you only need $O(\log|R|) = O(\log\log\Delta)$ steps. So can your time complexity improved to $O(\epsilon^{-2}k^6\log k\log\log\Delta)$?
>
> **Response:** For each guess $r \in R $, we run an instance of the algorithm for that specific $ r $. After each insertion or deletion, we update all these instances. As a result, the update time is affected by a factor of $ |R| = \log_{1+\epsilon} \Delta = O(\epsilon^{-1}\log{\Delta})$. We apologize for any confusion caused by the title of that paragraph and will revise it accordingly.
>
> **Question:** If I want to delete some point which is also a center of the solution. I think this center point should also be deleted since this point has been removed from this metric space. But it seems that Procedure 2 in the paper do not consider this situation. How do your algorithm handle the deletion of the center points?
>
> **Response:** As mentioned in the paper, we studied the non-discrete version of the problem, where centers can be any point in the metric space. Therefore, a center can be a deleted point. We would like to thank you for bringing to our attention the interesting question about the discrete version of the problem we study in the paper. Indeed, we have thought about this question during the short rebuttal period and have provide a proof that explains how our dynamic data structure can also provide a $6$-approximation solution for the discrete metric $k$-center problem with outliers, where centers must be selected from the input set of points.
> You can find the proof of this claim in the response to reviewer CFX4. We also attached a figure in the global rebuttal for a visual representation. Note that any feasible solution for the discrete version is also feasible for the non-discrete version. Therefore, the optimal radius for the discrete version is at least as large as that for the non-discrete version. This implies that, for some applications, the non-discrete version can offer more precise clustering. We believe both the discrete and non-discrete versions are important, each being better suited for different applications. Besides the improvement in approximation factor, another interesting aspect of our method is that it is **adversarially robust**, meaning it can handle an adversary who observes the computed solution after each update and adjusts the subsequent updates accordingly. In contrast, the dynamic $14$-approximation algorithm developed by Chan et al. is **not** adversarially robust. In particular, if their solution is revealed at any time, the adversary can impose an expensive update that requires time in $\Omega(n)$.
>
> **Question:** What is the size of your memory usage?
>
> **Response:** The space complexity of our algorithm is $O(\epsilon^{-1}\log(\Delta)n) $. This is because we maintain $ \epsilon^{-1} \log(\Delta) $ instances corresponding to different radius guesses, each containing up to $ n $ points. The space required to temporarily store the sample sets used in Algorithm $\texttt{OfflineCluster}$ is in $O(n)$.

---

### Official Review · Reviewer_CFX4 · 2024-07-12

**Soundness:** 4
**Presentation:** 3
**Contribution:** 3
**Rating:** 7
**Confidence:** 5

**Summary:**

The paper studies the k-center clustering problem with outliers in the fully dynamic setting. Specifically, given a metric space (M,d), in the (k,z)-clustering problem, the goal is to find at most k balls minimizing the maximum ball radius while excluding up to z points from the clustering. In the fully dynamic setting, the points are inserted or deleted from the underlying metric space, and the goal is to maintain the clustering faster than recomputing from scratch after each insertion or deletion.

The main contribution of this paper is a fully-dynamic, provable approximation algorithm for the (k,z) clustering problem that achieves (4+\eps)-approximation ratio, and can support point updates in O(\eps^{-2} k^6 \log k \log \Delta), with the caveat that the algorithm covers all but at most (1+\eps)z points (also known as a bi-criteria guarantee in the approximation algorithms literature). The algorithm is randomized, and works against an oblivious, online adversary. It also improves upon the approximation ratio of (14+\eps) achieved in the recent work Chan, Lattanzi, Sozio, and Wang [5] for the same problem, at the cost of increasing the running time by a factor of k^4.

From a technical point of view, the paper first introduces a static (4+\epsilon) approximation algorithm for the problem. The algorithm is itself based on the idea of successive random sampling, which proceeds as follows:

*  guess an optimal radius r (this is standard in (clustering) algorithms, and can be achieved by binary searching at the cost of paying (1+\eps) in the approx. factor, and a factor that is log of aspect ratio in the underlying metric space in the runtime);
* pick uniformly at random a subset of points S of the current point set, and pick a center c from S that covers a good fraction of the current points;
* grow a ball of radius 4r around the center c;
* remove the points belonging to the ball from the current point sets, and continue with the next iteration (level)
* the number of levels is bounded by roughly k
* the centers computed in each level are output as the solution the (k,z) clustering problem

The bulk of the effort is then to carefully set up a data-structure that maintains these levels (and the associated information) under point insertion and deletions. There are two main invariants: (i) the level invariant and (ii) the dense invariant (which is roughly controlling the number of points that remain unclustered in each level). The moral message of the paper is that when the invariants are broken at some level i, then you basically re-cluster everything from that level up.

**Strengths:**

*The k-center clustering is a fundamental problem in the clustering/unsupervised learning literature. The outlier version of the problem makes perfect sense, and it's of equal importance. The dynamic setting is very natural and has received increasing attention in the clustering literature in the last 5 years or so, with many papers trying to nail down the dynamic complexity of basic clustering objectives such as k-center, k-median and k-means. This work can be thought as a continuation of this line of work.

* The algorithm is simple and elegant (it should also be very implementable). Appropriate effort is put in explaining high level ideas. The paper reads well.

**Weaknesses:**

* Personally, I wouldn't agree that this is an improvement over the work of Chan et al. [5]. Indeed, it does achieve a better approximation ratio, but at the cost of increasing the runtime. The paper indeed provides a new, interesting trade-off, but I believe the abstract we should discuss this more carefully.

* Regarding techniques, while one can argue that idea of successive sampling is now folklore, one influential work that employs a very similar algorithm is due to Mettu-Plaxton "Optimal Time Bounds for Approximate Clustering". While the original algorithm is for k-median and k-means, there are follow up works that employ the same algorithm for k-center (albeit with worst approx guarantee than 2). I would encourage the authors to discuss differences/similarities between Mett-Plaxton and their work. In my opinion, this doesn't diminish the contribution of the paper at hand.

**Questions:**

Minor comments:

Lines 17-29, you should back up with citations the applications of clustering across many subfields of computer science and beyond
Line 36, 'various complications' sounds a bit odd (maybe challenges would be a better fit here)
Line 196, n_i, \alpha never defined before in text
Line 219, should your j start from 1 and not from i? -- similar comment for Definition 3.2

Major comment/question:
Line 146 -- can you please elaborate whether non-discrete version of the (k,z) clustering you study here is less challenging? I imagine this helps you a lot with deletions; even if you delete a point in P, you can still leave the underlying point in M serve as cluster center, which doesn't invoke a re-build. Does the work of Chan et al. [5] have the some restriction?

**Limitations:**

Yes,

---

> ### Author Rebuttal · Authors · 2024-08-07
>
> **Comment:** Personally, I wouldn't agree that this is an improvement over the work of Chan et al. [5]. Indeed, it does achieve a better approximation ratio, but at the cost of increasing the runtime. The paper indeed provides a new, interesting trade-off, but I believe the abstract we should discuss this more carefully.
>
> **Response:** Thank you for the comment. We will discuss this trade-off in more detail in the full version. Besides the improvement in approximation factor, another interesting aspect of our method is that it is **adversarially robust**, meaning it can handle an adversary who observes the computed solution after each update and adjusts the subsequent updates accordingly. In contrast, the dynamic $14$-approximation algorithm developed by Chan et al. is **not** adversarially robust. In particular, if their solution is revealed at any time, the adversary can impose an expensive update that requires time in $\Omega(n)$.
>
>
> **Comment:** Regarding techniques,[...] I would encourage the authors to discuss differences/similarities between Mett-Plaxton and their work. In my opinion, this doesn't diminish the contribution of the paper at hand.
>
> **Response:**  Thank you for highlighting the work of Mettu and Plaxton on 'Optimal Time Bounds for Approximate Clustering' and the subsequent research. We will be sure to discuss the differences and similarities between the work of Mettu and Plaxton and ours.
>
> **Question:** Lines 17-29, you should back up with citations the applications of clustering across many subfields of computer science and beyond
> Line 36, 'various complications' sounds a bit odd (maybe challenges would be a better fit here)
> Line 196, $n_i, \alpha$ never defined before in text
> Line 219, should your j start from 1 and not from i? -- similar comment for Definition 3.2
>
> **Response:** Thank you for the valuable suggestions. We will make sure to add more references and the mentioned definitions at the correct positions in the full version.
> The subset $P_i$ of points not covered in level $i$ is disjoint from the clusters $C_1,\ldots,C_{i-1}$. Hence, a clustering of $P_i$ does not need to contain these clusters. Similarly, a deletion of a point at level $i$ does not lead to a violation of the invariants in levels $1,\ldots,i-1$, but can potentially violate the invariants at higher levels. Therefore, we only need to recluster the levels from $i$ upwards. We will make sure to write this part more clearly in the full version.
>
> **Question:** Major comment/question: Line 146 -- can you ,[...] Does the work of Chan et al. [5] have the some restriction?
>
> **Response:** We would like to thank you for bringing to our attention the interesting question about the discrete version of the problem we study in the paper. Indeed, we have thought about this question during the short rebuttal period and determined how our dynamic data structure can also provide a $6$-approximation solution for the discrete metric $k$-center problem with outliers, where centers must be selected from the input set of points. Observe that any feasible solution for the discrete version is also valid for the non-discrete version. Therefore, the optimal radius in the discrete version is at least as large as that in the non-discrete version. This suggests that, in certain applications, the non-discrete version may provide more precise clustering. We believe that both the discrete and non-discrete versions are valuable, each being particularly suited to different applications. Chan et al. [5] consider the discrete version; however, they need to reconstruct the leveling after the deletion of a center, making their algorithm not adversarially robust.
>
> Here, we prove how our data structure can also offer a $6$-approximation solution for the discrete version. We also attached a figure in the global rebuttal for a visual representation.
>
> **Claim**: Our data structure can report a $6$-approximation solution for the discrete version of the $k$-center problem with $z$ outliers without increasing the time or space complexity.
>
> **Proof:** Let $r$ and $i$ be fixed, and $c_i$ be the center of cluster $C_i=B(c_i,4r)$ (Recall that $B(x,\rho)$ is the ball of radius $\rho$ centered at $x$). To provide a solution for the discrete version, we do the following.
>
> As long as $c_i$ is not deleted, we report $c_i$ as the $i$-th center. After the deletion of the point $c_i$, we consider two cases: $\min(z+1,\frac{n_i-z}{k-i+1}-\frac{\epsilon z}{\alpha k})\leq 0$ or $\min(z+1,\frac{n_i-z}{k-i+1}-\frac{\epsilon z}{\alpha k})>0$. For the first case we have $n_i<(1+\epsilon)z$ as $i\geq 1$ and $\alpha>1$. Hence, we can report all the $n_i$ points as outliers and stop. The dense invariant states that $|B(c_i,2r\cap P_i|\geq\min(z+1,\frac{n_i-z}{k-i+1}-\frac{\epsilon z}{\alpha k})$. Therefore, $|B(c_i,2r)\cap P_i|>0$ holds in the second case and there exists a point $\hat{p}\in B(c_i,2r)$. Then we report the point $\hat{p}\in B(c_i,2r)$ after the deletion of $c_i$.
>
> We next prove the $6$-approximation guarantee. To do this, we show that $C_i\subseteq B(\hat{p},6r)$. This implies that any feasible solution in our data structure with radius $4r$ for the non-discrete version can be used to report a feasible solution for the discrete version. Also, see attached the figure in the global rebuttal for a visual representation. Let $q\in C_i$. We show that $q\in B(\hat{p},6r)$. Since $q\in C_i=B(c_i,4r)$, we have $d(q,c_i)\leq 4r$ , where $d$ is the distance function. Moreover, $d(c_i \hat{p}) \leq 2r$ since $\hat{p}\in B(c_i,2r)$. Then by the triangle inequality we have $d(q, \hat{p}) \leq d(q,c_i)+d(c_i,\hat{p})\leq 4r+2r=6r.$
>
> The complexity of space and time remains to be discussed. To report a solution for the discrete version, it is enough to keep $B(c_i,2r)\cap P_i$. Note that our data structure already stores $C_i\cap P_i$ , and since $B(c_i,2r)\cap P_i \subseteq C_i\cap P_i$ , the space complexity and time complexity remain the same.

---

### Official Review · Reviewer_NQy9 · 2024-07-12

**Soundness:** 3
**Presentation:** 4
**Contribution:** 3
**Rating:** 7
**Confidence:** 4

**Summary:**

This paper studies the fully-dynamic $k$-center with outliers problem in the metric space. In this setting, operations (including insertion and deletion) appear over time. The performance evaluation of an algorithm is based on its cost approximation and the (amortized) update time. However, previous research has shown that when exactly excluding $z$ outliers is required, any $O(1)$-approximation algorithm incurs an $\Omega(z)$ update time. It is therefore reasonable to allow for a trade-off by permitting $(1+\varepsilon)z$ outliers in order to achieve efficient update times.

The proposed algorithm can be summarized as follows: It partitions the dataset into into at most $k+1$ levels, where each level $i$ represents a cluster $C_i$ with center $c_i$. The algorithm dynamically maintains a data structure that adapts to the operations performed in real-time. At every time, the data structure should maintain the following two invariants:

- level invariant: Each level $i$ has a cluster $C_i$ such that $P_{i+1} = P_i \backslash C_i$.
- dense invariant: For each center $c_i$ in $F_r$, $B(c_i, 2r)$ covers sufficient points in $P_i$.

The authors prove that it is a $(4+\varepsilon)$-approximation algorithm with $O(\varepsilon^{-2}k^6\log k\log \Delta)$ expected amortized update time.

**Strengths:**

- The paper is very well written and easy to follow.
- The algorithm provides a $(4+\varepsilon)$-approximation algorithm for the fully-dynamic $k$-center with $z$ outliers problem, which is an improvement over previous work.
- The paper includes a thorough theoretical analysis, proving the approximation guarantee and update time of the algorithm.

**Weaknesses:**

- the update time complexity $O(\varepsilon^{-2}k^6\log k\log \Delta)$ is not competitive, particularly for large $k$.
- The paper does not provide empirical results or practical evaluations of the algorithm, which could demonstrate its performance in real-world scenarios.

**Questions:**

- can this method be slightly modified to support the change of $k$ and $z$. Alternatively, is there any negative results on handling the dynamic $k$ and $z$.
- iDoes the analysis of the algorithm include a specific space complexity?  Additionally, i am curious about the hardness of the fully-dynamic clustering with outliers. Can you provide the lower bounds of the space and the update time in the context of the fully-dynamic setting?

**Limitations:**

thoretical paper

---

> ### Author Rebuttal · Authors · 2024-08-07
>
> **Comment:** the update time complexity $O(\varepsilon^{-2}k^6\log{k}\log{\Delta})$ is not competitive, particularly for large $k$.
>
> **Response:** Our goal was to develop a dynamic algorithm with a low approximation guarantee and a simple, elegant data structure. We did not focus on optimizing the exponent of $k$ in our running time. An interesting aspect of our method is that it is **adversarially robust**, meaning it can handle an adversary who observes the computed solution after each update and adjusts the subsequent updates accordingly. In contrast, the dynamic $14$-approximation algorithm developed by Chan et al. is not adversarially robust. If their solution is revealed at any time, the adversary can impose an expensive update that requires time in $\Omega(n)$.
>
> **Comment:** The paper does not provide empirical results or practical evaluations of the algorithm, which could demonstrate its performance in real-world scenarios.
>
> **Response:** We agree with the reviewer that providing empirical results or practical evaluations would be valuable and interesting work. Our main focus was to improve the $14$-approximation guarantee and determine how much we could refine the approximation factor, leading to the development of the first adversarially robust dynamic $4$-approximation algorithm for the metric $k$-center problem with outliers. We consider conducting experiments and benchmarking our dynamic algorithm against the algorithm by Chan et al. for both real and synthetic scenarios as future work.
>
> **Question:** Can this method be slightly modified to support the change of $k$ and $z$. Alternatively, is there any negative results on handling the dynamic $k$ and $z$.
>
> **Response:** Our data structure is designed with up to $k$ levels, based on known values of $k$ and $z$. We are unsure how to modify our data structure to handle changes in $k$ and $z$ during the execution of the algorithm. Investigating the development of such a dynamic algorithm would indeed be an interesting challenge.
>
> **Question:** Does the analysis of the algorithm include a specific space complexity? Additionally, i am curious about the hardness of the fully-dynamic clustering with outliers. Can you provide the lower bounds of the space and the update time in the context of the fully-dynamic setting?
>
> **Response:** The space complexity of our algorithm is $O(\epsilon^{-1}\log(\Delta)n)$. This is because we maintain $ \epsilon^{-1} \log(\Delta) $ instances corresponding to different radius guesses, each containing up to $ n $ points. For the sample sets needed in $\texttt{OfflineCluster}$, we also need $O(n)$ space because they form a subset of the dataset. Establishing a space lower bound for fully dynamic algorithms for $ k $-center with outliers would be an intriguing question.

---

> > ### Comment · Reviewer_NQy9 · 2024-08-12
> >
> > Thanks for the clarification. I will keep my original rating.

---

### Official Review · Reviewer_fRgM · 2024-07-16

**Soundness:** 4
**Presentation:** 3
**Contribution:** 2
**Rating:** 6
**Confidence:** 3

**Summary:**

The paper gives a new algorithm for the dynamic version of k-center clustering with outliers. The algorithm works in the fully dynamic model with both point insertions and deletions allowed. The points can belong to an arbitrary metric space, compared to some previous algorithms addressing low-dimensional metric spaces.

**Strengths:**

This is an important problem and the paper significantly improves the approximation factor.

**Weaknesses:**

The algorithm is randomized and works only against oblivious adversaries. This is a shared characteristic with the previous paper on this topic.

The algorithm achieves a better approximation factor than the previous algorithms, but it comes at the cost of significantly higher dependency in the update time on the number of clusters, k. In particular, the factor of $k^6$ is probably prohibitive in practice.

Overall, even though this is a solid contribution to the study of the problem, this is not a breakthrough.

I also wish the authors did a better job highlighting the technical ideas that lead to the improvement over the previous algorithm.

**Questions:**

Is there a good reason why the algorithm is randomized? What are the obstacles to obtaining deterministic algorithms in this model?

The factor of $k^6$ in your upper bound is large. Is there a reason why in practice the algorithm would be significantly less costly?

**Limitations:**

No specific societal limitations to address. This is a theoretical work concerning a traditional computational problem.

---

> ### Author Rebuttal · Authors · 2024-08-07
>
> **Comment:** The algorithm is randomized and works only against oblivious adversaries. This is a shared characteristic with the previous paper on this topic.
>
> **Question:** Is there a good reason why the algorithm is randomized? What are the obstacles to obtaining deterministic algorithms in this model?
>
> **Response:** We address these two points together in our response. Interestingly, we have discovered that our dynamic algorithm not only can handle oblivious adversaries but is even adversarially robust. A dynamic algorithm is called adversarially robust if its performance guarantees (including the approximation factor and update time) are preserved even when the sequence of updates (i.e., insertions and deletions) is adaptively chosen by an adversary who observes the outputs of our algorithm throughout the sequence and adjusts them accordingly. An adversarially robust dynamic algorithm is more powerful than one designed to handle only an oblivious adversary, as an oblivious adversary cannot access the algorithm's outputs after each update. In particular, we only use random bits to sample from large clusters at each level and maintain a solution. (This is the only component of our algorithm that uses randomness.) The adversary, however, can observe the maintained solution after each update and adaptively modify future updates (insertions and deletions) accordingly. In contrast, the dynamic $14$-approximation algorithm developed by Chan et al. is designed only for oblivious adversaries and is therefore not adversarially robust. In particular, if their solution were revealed at any time, the adversary could impose costly update time  $\Omega(n)$.
>
> **Question:** The factor of $k^6$ in your upper bound is large. Is there a reason why in practice the algorithm would be significantly less costly?
>
> **Response:** Thank you for raising this interesting question. The worst case occurs when the gap between the size of the ball chosen in the offline algorithm and the threshold is negligible, but we believe this is not always the case in practice. To increase this gap, we propose selecting the ball of maximum size in Line 8 of the OfflineCluster procedure. This adjustment does not change the theoretical bounds but can make the algorithm faster in practice.

---

> > ### Comment · Reviewer_fRgM · 2024-08-12
> >
> > Thank you for your response!
> >
> > > we have discovered that our dynamic algorithm not only can handle oblivious adversaries but is even adversarially robust
> >
> > This is very interesting and if true, it definitely makes your algorithm more interesting. I'm very curious about this but I'm not sure I'll have time to reread your paper to see if I have any doubts about this. Whatever happens to your paper, it would of course be great to explicitly address this in any future version of the paper.

---

### Author Rebuttal · Authors · 2024-08-07

Thank you to the reviewers for their insightful comments and valuable feedback. We also appreciate the time they dedicated to reviewing our work.

Here, we also provide a file with a figure illustrating our responses to reviewers CFX4 and fWP4.

---

### Decision · Program_Chairs · 2024-09-25

**Decision:**

Accept (poster)

**Comment:**

This paper gives new ratio-time trade-offs for k-center with outliers in a general metric space. The reviews are overall positive, and the main strength is the improved ratio (albeit the dependence on k in running time is slightly worse). The main weakness is that the paper does not seem to make significant progress in terms of techniques, and there is no experiment.

During the discussion, the authors claim their result also works against an adaptive adversary (which was not mentioned in the submitted version). This feature is important and could be a major strength of the paper. The authors should include the relevant claim/proof to the final version.